# The *Aquilegia* genome provides insight into adaptive radiation and reveals an extraordinarily polymorphic chromosome with a unique history

Danièle L Filiault[1], Evangeline S Ballerini[2], Terezie Mandáková[3], Gökçe Aköz[1,4], Nathan J Derieg[2], Jeremy Schmutz[5,6], Jerry Jenkins[5,6], Jane Grimwood[5,6], Shengqiang Shu[5], Richard D Hayes[5], Uffe Hellsten[5], Kerrie Barry[5], Juying Yan[5], Sirma Mihaltcheva[5], Miroslava Karafiátová[7], Viktoria Nizhynska[1], Elena M Kramer[8], Martin A Lysak[3], Scott A Hodges[2]*, Magnus Nordborg[1]*

[1]Gregor Mendel Institute, Austrian Academy of Sciences, Vienna BioCenter, Vienna, Austria; [2]Department of Ecology, Evolution and Marine Biology, University of California, Santa Barbara, United States; [3]Central-European Institute of Technology, Masaryk University, Brno, Czech Republic; [4]Vienna Graduate School of Population Genetics, Vienna, Austria; [5]Department of Energy, Joint Genome Institute, Walnut Creek, United States; [6]HudsonAlpha Institute of Biotechnology, Alabama, United States; [7]Institute of Experimental Botany, Centre of the Region Haná for Biotechnological and Agricultural Research, Olomouc, Czech Republic; [8]Department of Organismic and Evolutionary Biology, Harvard University, Cambridge, United States

*For correspondence:
scott.hodges@lifesci.ucsb.edu (SAH);
magnus.nordborg@gmi.oeaw.ac.at (MN)

**Abstract** The columbine genus *Aquilegia* is a classic example of an adaptive radiation, involving a wide variety of pollinators and habitats. Here we present the genome assembly of *A. coerulea* 'Goldsmith', complemented by high-coverage sequencing data from 10 wild species covering the world-wide distribution. Our analyses reveal extensive allele sharing among species and demonstrate that introgression and selection played a role in the *Aquilegia* radiation. We also present the remarkable discovery that the evolutionary history of an entire chromosome differs from that of the rest of the genome – a phenomenon that we do not fully understand, but which highlights the need to consider chromosomes in an evolutionary context.
DOI: https://doi.org/10.7554/eLife.36426.001

## Introduction

Understanding adaptive radiation is a longstanding goal of evolutionary biology (*Schluter, 2000*). As a classic example of adaptive radiation, the *Aquilegia* genus has outstanding potential as a subject of such evolutionary studies (*Hodges et al., 2004*; *Hodges and Derieg, 2009*; *Kramer, 2009*). The genus is made up of about 70 species distributed across Asia, North America, and Europe (*Munz, 1946*) (*Figure 1*). Distributions of many *Aquilegia* species overlap or adjoin one another, sometimes forming notable hybrid zones (*Grant, 1952*; *Hodges and Arnold, 1994b*; *Li et al., 2014*). Additionally, species tend to be widely interfertile, especially within geographic regions (*Taylor, 1967*).

Phylogenetic studies have defined two concurrent, yet contrasting, adaptive radiations in *Aquilegia* (*Bastida et al., 2010*; *Fior et al., 2013*). From a common ancestor in Asia, one radiation

**Figure 1.** Distribution of *Aquilegia* species. There are ~70 species in the genus *Aquilegia*, broadly distributed across temperate regions of the Northern Hemisphere (grey). The 10 *Aquilegia* species sequenced here were chosen as representatives spanning this geographic distribution as well as the diversity in ecological habitat and pollinator-influenced floral morphology of the genus. *Semiaquilegia adoxoides*, generally thought to be the sister taxon to *Aquilegia* (**Fior et al., 2013**), was also sequenced. A representative photo of each species is shown and is linked to its approximate distribution.

DOI: https://doi.org/10.7554/eLife.36426.002

The following figure supplement is available for figure 1:

**Figure supplement 1.** Origin of species samples used for sequencing.

DOI: https://doi.org/10.7554/eLife.36426.003

occurred in North America via Northeastern Asian precursors, while a separate Eurasian radiation took place in central and western Asia and Europe. While adaptation to different habitats is thought to be a common force driving both radiations, shifts in primary pollinators also play a substantial role in North America (**Whittall and Hodges, 2007**; **Bastida et al., 2010**). Previous phylogenetic studies have frequently revealed polytomies (**Hodges and Arnold, 1994b**; **Ro et al., 1997**; **Whittall and Hodges, 2007**; **Bastida et al., 2010**; **Fior et al., 2013**), suggesting that many *Aquilegia* species are very closely related.

Genomic data are beginning to uncover the extent to which interspecific variant sharing reflects a lack of strictly bifurcating species relationships, particularly in the case of adaptive radiation.

Discordance between gene and species trees has been widely observed (*Novikova et al., 2016* and references 15, 34–44 therein; *Svardal et al., 2017*; *Malinsky et al., 2017*), and while disagreement at the level of individual genes is expected under standard population genetics coalescent models (*Takahata, 1989*) (also known as 'incomplete lineage sorting' [*Avise and Robinson, 2008*]), there is increased evidence for systematic discrepancies that can only be explained by some form of gene flow (*Green et al., 2010*; *Novikova et al., 2016*; *Svardal et al., 2017*; *Malinsky et al., 2017*). The importance of admixture as a source of adaptive genetic variation has also become more evident (*Lamichhaney et al., 2015*; *Mallet et al., 2016*; *Pease et al., 2016*). Hence, rather than being a problem to overcome in phylogenetic analysis, non-bifurcating species relationships could actually describe evolutionary processes that are fundamental to understanding speciation itself. Here we generate an *Aquilegia* reference genome based on the horticultural cultivar *Aquilegia coerulea* 'Goldsmith' and perform resequencing and population genetic analysis of 10 additional individuals representing North American, Asian, and European species, focusing in particular on the relationship between species.

## Results

### Genome assembly and annotation

We sequenced an inbred horticultural cultivar (*A. coerulea* 'Goldsmith') using a whole genome shotgun sequencing strategy. A total of 4,773,210 Sanger sequencing reads from seven genomic libraries (*Supplementary file 1*) were assembled to generate 2529 scaffolds with an N50 of 3.1 Mbp (*Supplementary file 2*). With the aid of two genetic maps, we assembled these initial scaffolds into a 291.7 Mbp reference genome consisting of 7 chromosomes (282.6 Mbp) and an additional 1027 unplaced scaffolds (9.13 Mbp) (*Supplementary file 3*). The completeness of the assembly was validated using 81,617 full length cDNAs from a variety of tissues and developmental stages (*Kramer and Hodges, 2010*), of which 98.69% mapped to the assembly. We also assessed assembly accuracy using Sanger sequencing of 23 full-length BAC clones. Of more than 3 million base pairs sequenced, only 1831 were found to be discrepant between BAC clones and the assembled reference (*Supplementary file 4*). To annotate genes in the assembly, we used RNAseq data generated from a variety of tissues and *Aquilegia* species (*Supplementary file 5*), EST data sets (*Kramer and Hodges, 2010*), and protein homology support, yielding 30,023 loci and 13,527 alternate transcripts. The *A. coerulea* 'Goldsmith' v3.1 genome release is available on Phytozome (https://phytozome.jgi.doe.gov/). For a detailed description of assembly and annotation, see Materials and methods.

### Polymorphism and divergence

We deeply resequenced one individual from each of ten *Aquilegia* species (*Figure 1* and *Figure 1—figure supplement 1*). Sequences were aligned to the *A. coerulea* 'Goldsmith' v3.1 reference using bwa-mem (*Li and Durbin, 2009*; *Li, 2013*) and variants were called using GATK Haplotype Caller (*McKenna et al., 2010*). Genomic positions were conservatively filtered to identify the portion of the genome in which variants could be reliably called across all ten species (see Materials and methods for alignment, SNP calling, and genome filtration details). The resulting callable portion of the genome was heavily biased towards genes and included 57% of annotated coding regions (48% of gene models), but only 21% of the reference genome as a whole.

Using these callable sites, we calculated nucleotide diversity as the percentage of pairwise sequence differences in each individual. Assuming random mating, this metric reflects both individual sample heterozygosity and nucleotide diversity in the species as a whole. Of the ten individuals, most had a nucleotide diversity of 0.2–0.35% (*Figure 2a*), similar to previous estimates of nucleotide diversity in *Aquilegia* (*Cooper et al., 2010*), yet lower than that of a typical outcrossing species (*Leffler et al., 2012*). While likely partially attributable to enrichment for highly conserved genomic regions with our stringent filtration, this atypically low nucleotide diversity could also reflect inbreeding. Additionally, four individuals in our panel had extended stretches of near-homozygosity (defined as nucleotide diversity <0.1%) consistent with recent inbreeding (*Figure 2—figure supplement 1*). *Aquilegia* has no known self-incompatibility mechanism, and selfing does appear to be common. However, inbreeding in adult plants is generally low, suggesting substantial inbreeding depression (*Montalvo, 1994*; *Herlihy and Eckert, 2002*; *Yang and Hodges, 2010*).

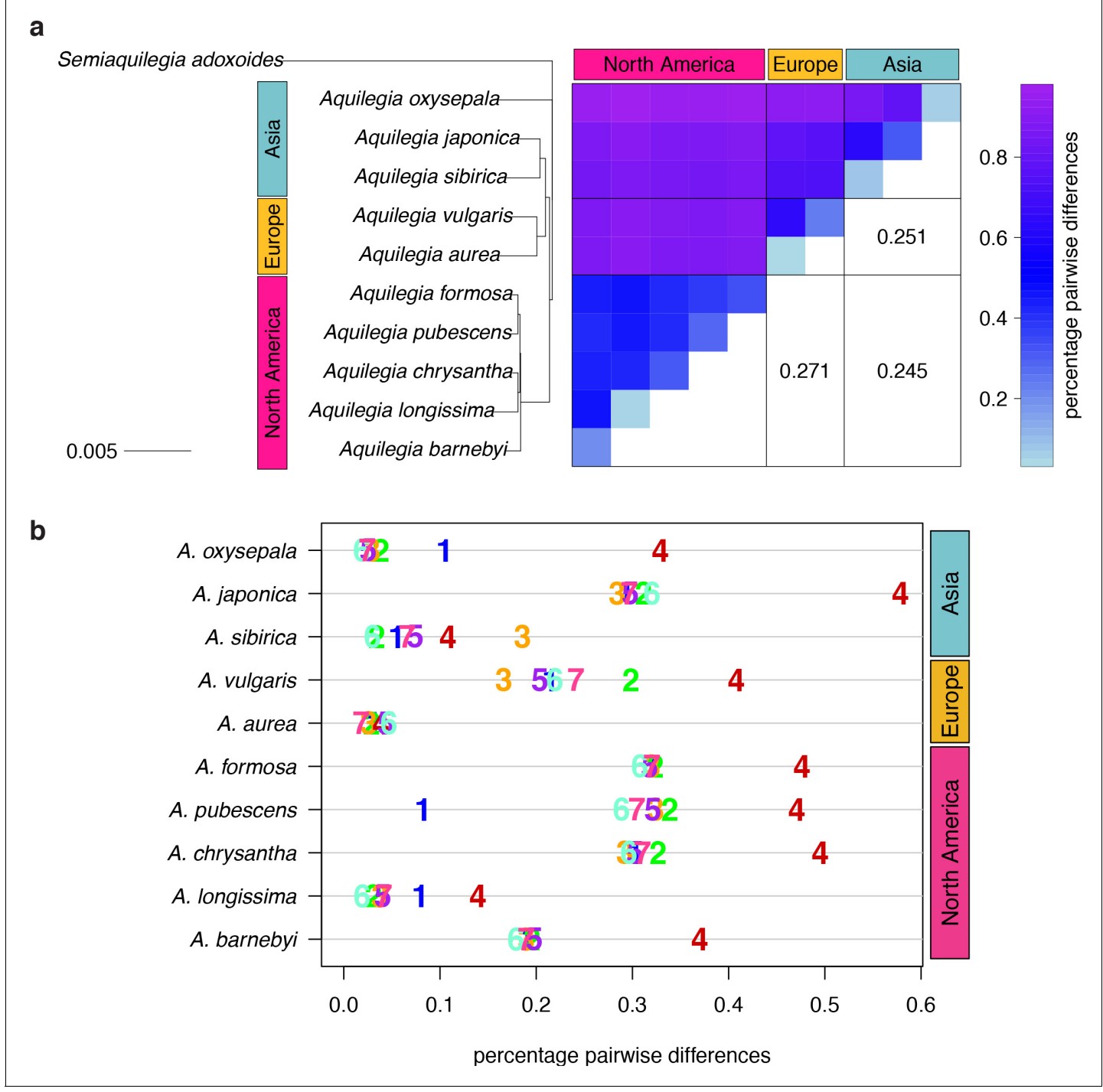

**Figure 2.** Polymorphism and divergence in *Aquilegia*. (**a**) The percentage of pairwise differences within each species (estimated from individual heterozygosity) and between species (divergence). $F_{ST}$ values between geographic regions are given on the lower half of the pairwise differences heatmap. Both heatmap axes are ordered according to the neighbor joining tree to the left. This tree was constructed from a concatenated data set of reliably-called genomic positions. (**b**) Polymorphism within each sample by chromosome. Per-chromosome values are indicated by the chromosome number.

DOI: https://doi.org/10.7554/eLife.36426.004

The following figure supplements are available for figure 2:

**Figure supplement 1.** Polymorphism across the genome in all ten species samples.

DOI: https://doi.org/10.7554/eLife.36426.005

**Figure supplement 2.** Species and chromosome trees of *Aquilegia*.

*Figure 2 continued on next page*

*Figure 2 continued*

DOI: https://doi.org/10.7554/eLife.36426.006

We next considered nucleotide diversity between individuals as a measure of species divergence. Species divergence within a geographic region (0.38–0.86%) was often only slightly higher than within-species diversity, implying extensive variant sharing, while divergence between species from different geographic regions was markedly higher (0.81–0.97%; *Figure 2a*). $F_{ST}$ between geographic regions (0.245–0.271) was similar to that between outcrossing species of the *Arabidopsis* genus (*Novikova et al., 2016*), yet lower than between most vervet species pairs (*Svardal et al., 2017*), and higher than between cichlid groups in Malawi (*Loh et al., 2013*) or human ethnic groups (*McVean et al., 2012*). The topology of trees constructed with concatenated genome data (neighbor joining (*Figure 2a*), RAxML (*Figure 2—figure supplement 2a*)) were in broad agreement with previous *Aquilegia* phylogenies (*Hodges and Arnold, 1994a*; *Ro and McPheron, 1997*; *Whittall and Hodges, 2007*; *Bastida et al., 2010*; *Fior et al., 2013*), with one exception: while *A. oxysepala* is sister to all other *Aquilegia* species in our analysis, it had been placed within the large Eurasian clade with moderate to strong support in previous studies (*Bastida et al., 2010*; *Fior et al., 2013*).

Surprisingly, levels of polymorphism were generally strikingly higher on chromosome four (*Figure 2b*). Exceptions were apparently due to inbreeding, especially in the case of the *A. aurea* individual, which appears to be almost completely homozygous (*Figure 2a* and *Figure 2—figure supplement 1*). The increased polymorphism on chromosome four is only partly reflected in increased divergence to an outgroup species (*Semiaquilegia adoxoides*), suggesting that it represents deeper coalescence times rather than simply a higher mutation rate (mean ratio chromosome four/genome at fourfold degenerate sites: polymorphism = 2.258, divergence = 1.201, *Supplementary file 6*).

## Discordance between gene and species trees

To assess discordance between gene and species (genome) trees, we constructed a cloudogram of trees drawn from 100 kb windows across the genome (*Figure 3a*). Fewer than 1% of these window-based trees were topologically identical to the species tree. North American species were consistently separated from all others (96% of window trees) and European species were also clearly delineated (67% of window trees). However, three bifurcations delineating Asian species were much less common: the *A. japonica* and *A. sibirica* sister relationship (45% of window trees), *A. oxysepala* as sister to all other species (30% of window trees), and the split demarcating the Eurasian radiation (31% of window trees). These results demonstrate a marked discordance of gene and species trees throughout both *Aquilegia* radiations.

The gene tree analysis also highlighted the unique evolutionary history of chromosome four. Of 217 unique subtrees observed in gene trees, nine varied significantly in frequency between chromosomes (chi-square test p-value < 0.05 after Bonferroni correction; *Figure 3b–d* and *Figure 3—figure supplements 1* and *2*). Trees describing a sister species relationship between *A. pubescens* and *A. barnebyi* were more common on chromosome one, but chromosome four stood out with respect to eight other relationships, most of them related to *A. oxysepala* (*Figure 3d*). Although *A. oxysepala* was sister to all other species in our genome tree, the topology of the chromosome four tree was consistent with previously-published phylogenies in that it placed *A. oxysepala* within the Eurasian clade (*Bastida et al., 2010*; *Fior et al., 2013*) (*Figure 2—figure supplement 2b,c*). Subtree prevalences were in accordance with this topological variation (*Figure 3b–d*). The subtree delineating all North American species was also less frequent on chromosome four, indicating that the history of the chromosome is discordant in both radiations. We detected no patterns in the prevalence of any chromosome-discordant subtree that would suggest structural variation or a large introgression (*Figure 3—figure supplement 3*).

## Polymorphism sharing across the genus

We next polarized variants against an outgroup species (*S. adoxoides*) to explore the prevalence and depth of polymorphism sharing. Private derived variants accounted for only 21–25% of

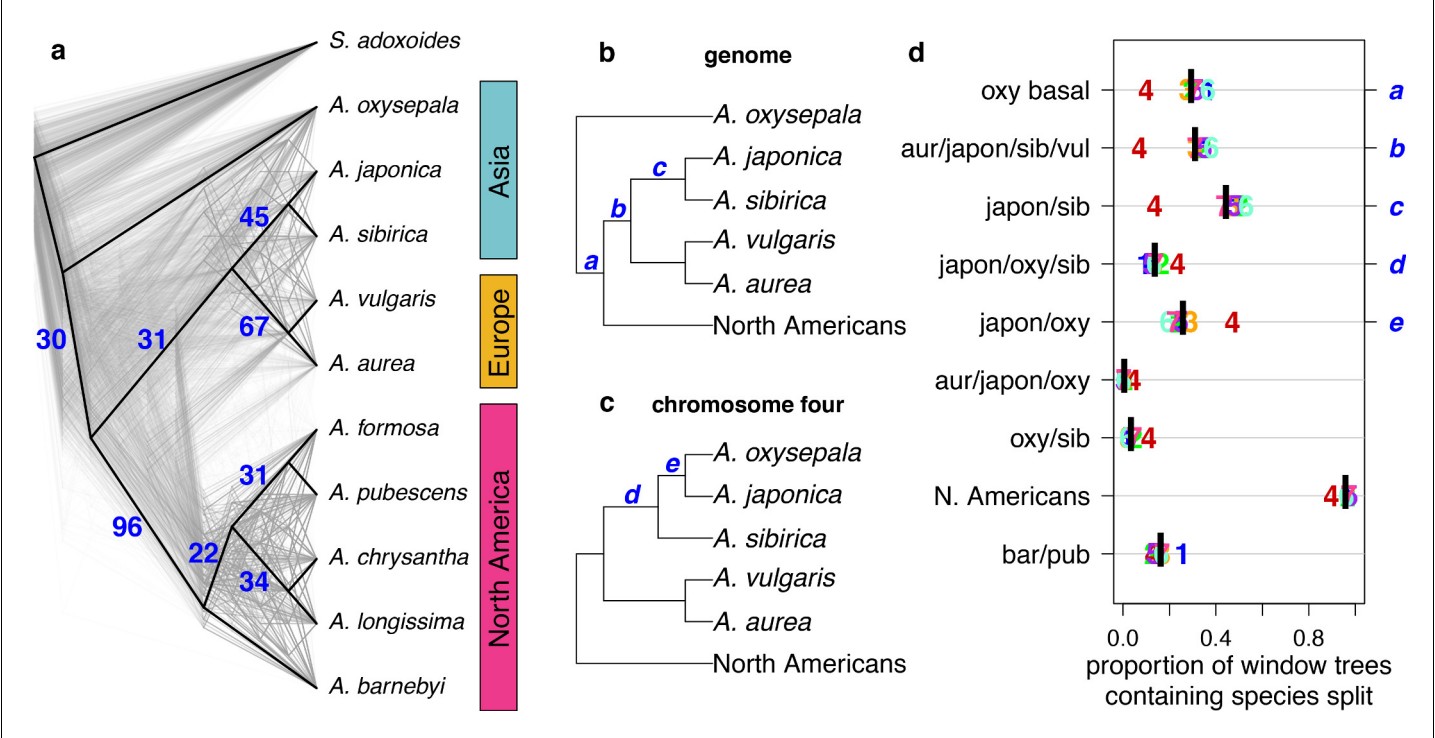

**Figure 3.** Discordance between gene and species trees. (a) Cloudogram of neighbor joining (NJ) trees constructed in 100 kb windows across the genome. The topology of each window-based tree is co-plotted in grey and the whole genome NJ tree shown in *Figure 2a* is superimposed in black. Blue numbers indicate the percentage of window trees that contain each of the subtrees observed in the whole genome tree. (b) Genome NJ tree topology. Blue letters a-c on the tree denote subtrees a-c in panel (d). (c) Chromosome four NJ tree topology. Blue letters d and e on the tree denote subtrees d and e in panel (d). (d) Prevalence of each subtree that varied significantly by chromosome. Genomic (black bar) and per chromosome (chromosome number) values are given.

DOI: https://doi.org/10.7554/eLife.36426.007

The following figure supplements are available for figure 3:

**Figure supplement 1.** Proportion of significantly-varying subtrees by chromosome.
DOI: https://doi.org/10.7554/eLife.36426.008
**Figure supplement 2.** *P*-values of proportion tests by chromosome for significantly-different trees.
DOI: https://doi.org/10.7554/eLife.36426.009
**Figure supplement 3.** Subtree prevalence across chromosomes for the nine significantly-different subtrees.
DOI: https://doi.org/10.7554/eLife.36426.010

polymorphic sites in North American species and 36–47% of variants in Eurasian species (*Figure 4a*). The depth of polymorphism sharing reflected the two geographically-distinct radiations. North American species shared 34–38% of their derived variants within North America, while variants in European and Asian species were commonly shared across two geographic regions (18–22% of polymorphisms, predominantly shared between Europe and Asia; *Figure 4b,c*; *Figure 4—figure supplement 1*). Strikingly, a large percentage of derived variants occurred in all three geographic regions (22–32% of polymorphisms, *Figure 4d*), demonstrating that polymorphism sharing in *Aquilegia* is extensive and deep.

In all species examined, the proportion of deeply shared variants was higher on chromosome four (*Figure 4d*), largely due to a reduction in private variants, although sharing at other depths was also reduced in some species. Variant sharing on chromosome four within Asia was higher in both *A. oxysepala* and *A. japonica* (*Figure 4b*), primarily reflecting higher variant sharing between these species (Figure 6a).

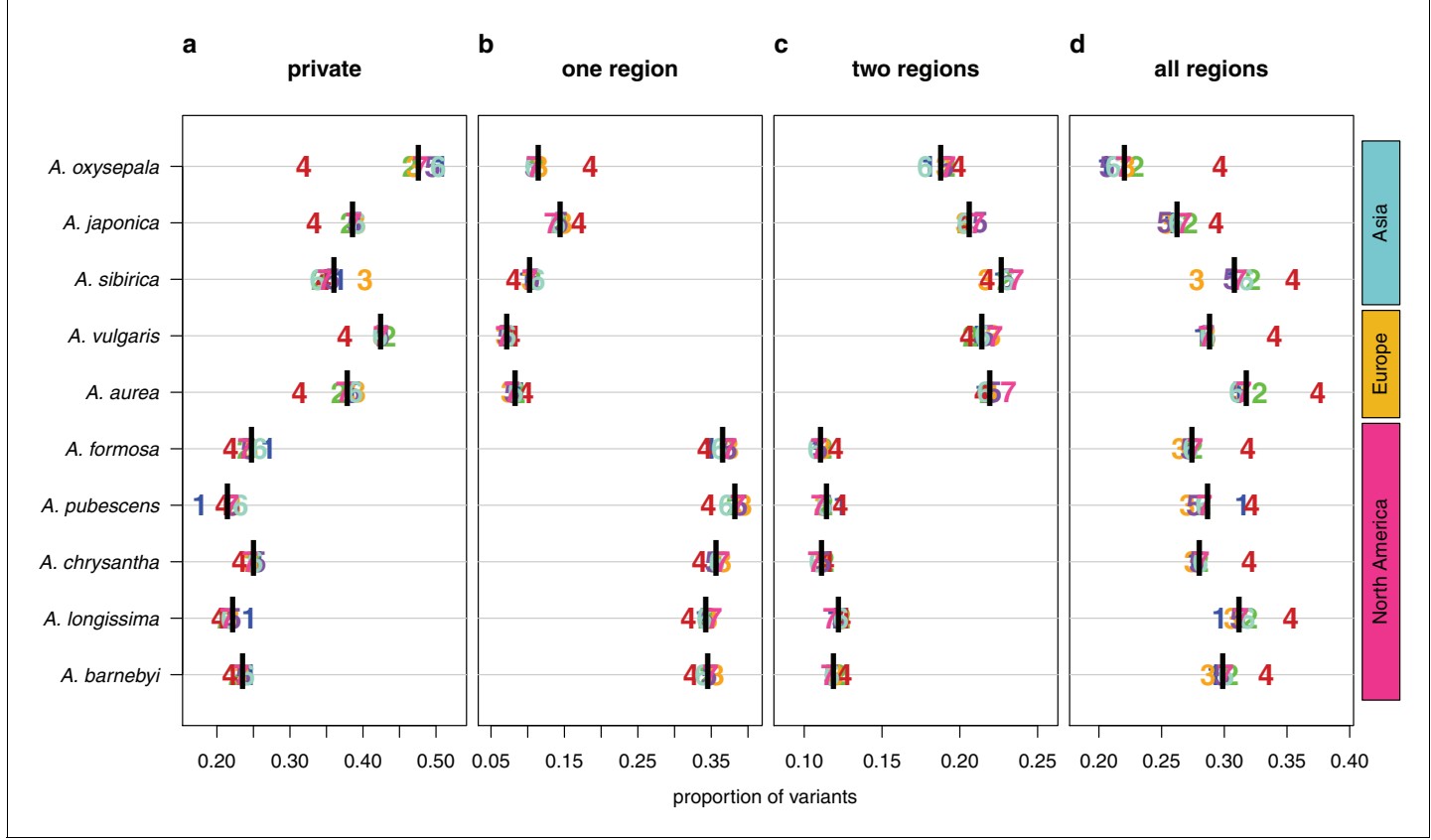

**Figure 4.** Sharing patterns of derived polymorphisms. Proportion of derived variants (**a**) private to an individual species, (**b**) shared within the geographic region of origin, (**c**) shared across two geographic regions, and (**d**) shared across all three geographic regions. Genomic (black bar) and chromosome (chromosome number) values, for all 10 species.

DOI: https://doi.org/10.7554/eLife.36426.011

The following figure supplement is available for figure 4:

**Figure supplement 1.** Sharing pattern percentages by pattern type.

DOI: https://doi.org/10.7554/eLife.36426.012

## Evidence of gene flow

Consider three species, H1, H2, and H3. If H1 and H2 are sister species relative to H3, then, in the absence of gene flow, H3 must be equally related to H1 and H2. The D statistic (*Green et al., 2010*; *Durand et al., 2011*) tests this hypothesis by comparing the number of derived variants shared between H3, and H1 and H2, respectively. A non-zero D statistic reflects an asymmetric pattern of allele sharing, implying gene flow between H3 and one of the two sister species, that is that speciation was not accompanied by complete reproductive isolation. If *Aquilegia* diversification occurred via a series of bifurcating species splits characterized by reproductive isolation, bifurcations in the species tree should represent combinations of sister and outgroup species with symmetric allele sharing patterns (D = 0). Given the high discordance of gene and species trees at the individual species level, we focused on testing a simplified tree topology based on the three groups whose bifurcation order seemed clear: (1) North American species, (2) European species, and (3) Asian species not including *A. oxysepala*. In all tests, *S. adoxoides* was used to determine the ancestral state of alleles.

We first tested each North American species as H3 against all combinations of European and Asian (without *A. oxysepala*) species as H1 and H2 (*Figure 5a–c*). As predicted, the North American split was closest to resembling speciation with strict reproductive isolation, with little asymmetry in allele sharing between North American and Asian species and low, but significant, asymmetry

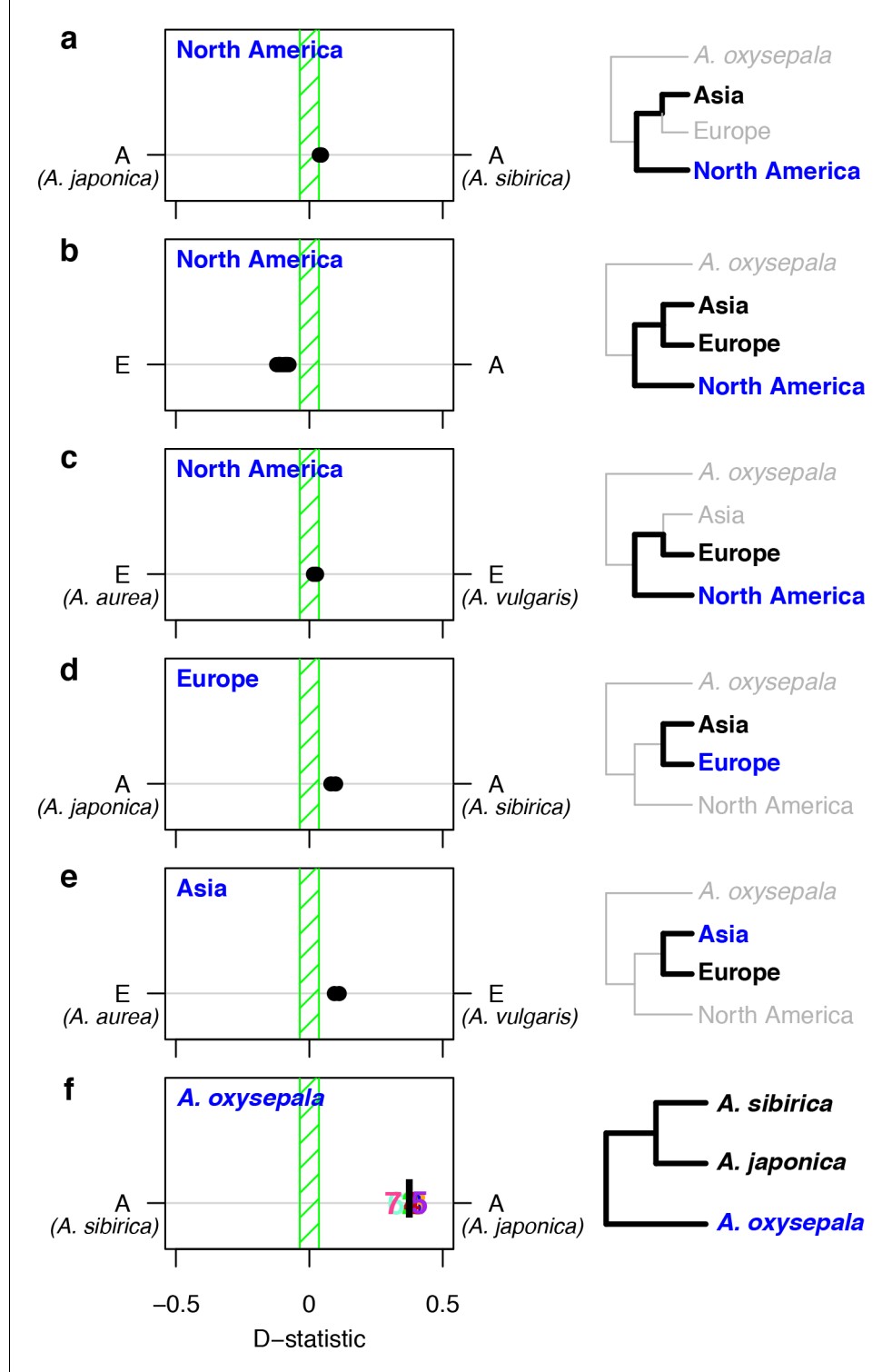

**Figure 5.** D statistics demonstrate gene flow during *Aquilegia* speciation. D statistics for tests with (**a–c**) all North American species, (**d**) both European species, (**e**) Asian species other than *A. oxysepala*, and (**f**) *A. oxysepala* as H3 species. All tests use *S. adoxoides* as the outgroup. D statistics outside the green shaded areas are significantly different from zero. In (**a–e**), each individual dot represents the D statistic for a test done with a unique species combination. In (**f**), D statistics are presented by chromosome (chromosome number) or by the genome-wide value (black bar). In all panels, E = European and A = Asian without *A. oxysepala*. In some cases, individual species names are given when the geographical region designation consists of a single species. Right hand panels are a
*Figure 5 continued on next page*

*Figure 5 continued*

graphical representation of the D statistic tests in the corresponding left hand panels. Trees are a simplified version of the genome tree topology (*Figure 2b*), in which the bold sub tree(s) represent the bifurcation considered in each set of tests. H3 species are noted in blue while the H1 and H2 species are specified in black. (*Figure 5—source data 1*).

DOI: https://doi.org/10.7554/eLife.36426.013

The following source data is available for figure 5:

**Source data 1.** (D statistics).

DOI: https://doi.org/10.7554/eLife.36426.014

between North American and European species (*Figure 5b*). Next, we considered allele sharing between European and Asian (without *A. oxysepala*) species (*Figure 5d,e*). Here we found non-zero D statistics for all species combinations. Interestingly, the patterns of asymmetry between these two regions were reticulate: Asian species shared more variants with the European *A. vulgaris* while European species shared more derived alleles with the Asian *A. sibirica*. D statistics therefore demonstrate widespread asymmetry in variant sharing between *Aquilegia* species, suggesting that speciation processes throughout the genus were not characterized by strict reproductive isolation.

Although non-zero D statistics are usually interpreted as being due to gene flow in the form of admixture between species, they can also result from gene flow between incipient species. Either way, speciation precedes reproductive isolation. The possibility that different levels of purifying selection in H1 or H2 explain the observed D statistics can probably be ruled out, since D statistics do not differ when calculated with only fourfold degenerate sites (*p*-value < 2.2 x 10$^{-16}$, adjusted $R^2$ = 0.9942, *Figure 5—source data 1*). Non-zero D statistics could also indicate that the bifurcation order tested was incorrect, but even tests based on alternative tree topologies resulted in few D statistics that equal zero (*Figure 5—source data 1*). Therefore, the non-zero D statistics observed in *Aquilegia* most likely reflect a pattern of reticulate evolution throughout the genus.

Since variant sharing between *A. oxysepala* and *A. japonica* was higher on chromosome four (*Figure 6a*), and hybridization between these species has been reported (*Li et al., 2014*) we wondered whether gene flow could explain the discordant placement of *A. oxysepala* between chromosome four and genome trees (*Figure 3b,c*). Indeed, when the genome tree was taken as the bifurcation order, D statistics were elevated between these species (*Figure 5f*). A relatively simple coalescent model allowing for bidirectional gene flow between *A. oxysepala* and *A. japonica* (*Figure 6b*) demonstrated that doubling the population size (N) to reflect chromosome four's polymorphism level (i.e. halving the coalescence rate) could indeed shift tree topology proportions (*Figure 6c*, row 2). However, recreating the observed allele sharing ratios on chromosome four (*Figure 6a*) required some combination of increased migration (m) and/or N (*Figure 6c*, rows 3–4). It is plausible that gene flow might differentially affect chromosome four, and we will return to this topic in the next section. Although the similarity of the D statistic across chromosomes (*Figure 5f*) might seem inconsistent with increased migration on chromosome four, the D statistic reaches a plateau in our simulations such that many different combinations of m and N produce similar D values (*Figure 6c* and *Figure 6—figure supplement 1*). Therefore, an increase in migration rate and deeper coalescence can explain the tree topology of chromosome four, a result that might explain inconsistencies in *A. oxysepala* placement in previous phylogenetic studies (*Bastida et al., 2010*; *Fior et al., 2013*).

## The pattern of polymorphism on chromosome four

In most of the sequenced *Aquilegia* species, the level of polymorphism on chromosome four is twice as high as in the rest of the genome (*Figure 2b*). This unique pattern could be: (1) an artifact of biases in polymorphism detection between chromosomes, (2) the result of a higher neutral mutation rate on chromosome four, or (3) the result of deeper coalescence times on chromosome four (allowing more time for polymorphism to accumulate).

While it is impossible to completely rule out phenomena such as cryptic copy number variants (CNV), for the pattern to be entirely attributable to artefacts would require that half of the polymorphism on chromosome four be spurious. This scenario is extremely unlikely given the robustness of

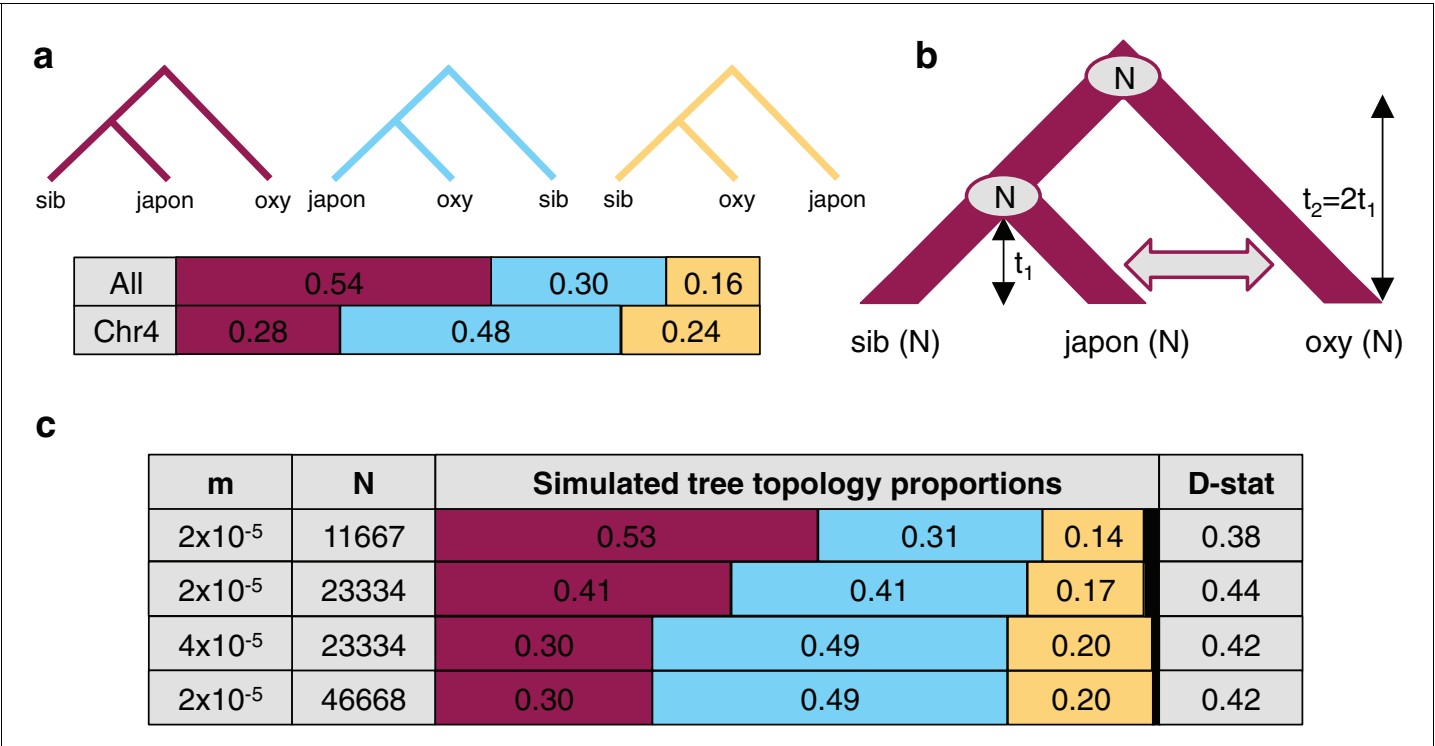

**Figure 6.** The effect of differences in coalescence time and gene flow on tree topologies. (**a**) The observed proportion of informative derived variants supporting each possible Asian tree topology genome-wide and on chromosome four. Species considered include *A. oxysepala* (oxy), *A. japonica* (japon), and *A. sibirica* (sib). (**b**) The coalescent model with bidirectional gene flow in which *A. oxysepala* diverges first at time $t_2$, but later hybridizes with *A. japonica* between t = 0 and t1 at a rate determined by per-generation migration rate, m. The population size (**N**) remains constant at all times. (**c**) The proportion of each tree topology and estimated D statistic for simulations using four combinations of m and Nvalues (t1 = 1 in units of N generations). The combination presented in the first row (m = $2x10^{-5}$ and N = 11667) generates tree topology proportions that match observed allele sharing proportions genomewide. Simulations with increased m and/or N (rows 3–4) result in proportions which more closely resemble those observed for chromosome four. Colors in proportion plots refer to tree topologies in (**a**), with black bars representing the residual probability of seeing no coalescence event. While this simulation assumes symmetric gene flow, similar results were seen for models incorporating both unidirectional and asymmetric gene flow (***Figure 6—figure supplements 1*** and ***2***).

DOI: https://doi.org/10.7554/eLife.36426.015

The following figure supplements are available for figure 6:

**Figure supplement 1.** Model output for all three gene flow scenarios.
DOI: https://doi.org/10.7554/eLife.36426.016
**Figure supplement 2.** Tree topology proportions simulated under assymmetric and unidirectional models.
DOI: https://doi.org/10.7554/eLife.36426.017

the result to a variety of CNV detection methods (***Supplementary file 7***). Similarly, the pattern cannot wholly be explained by a higher neutral mutation rate. If this were the case, both divergence and polymorphism would be elevated to the same extent on chromosome four (***Kimura, 1983***). As noted above, this not the case (***Supplementary file 6***). Thus the higher level of polymorphism on chromosome four must to some extent reflect differences in coalescence time, which can only be due to selection.

Although it is clear that selection can have a dramatic effect on the history of a single locus, the chromosome-wide pattern we observe (***Figure 2—figure supplement 1***) is difficult to explain. Chromosome four recombines freely (***Figure 7a***), suggesting that polymorphism is not due to selection on a limited number of linked loci, such as might be observed if driven by an inversion or large supergene. Selection must thus be acting on a very large number of loci across the chromosome.

Balancing selection is known to elevate polymorphism, and in a number of plant species, disease resistance (*R*) genes show signatures of balancing selection (***Karasov et al., 2014***). While such signatures have not yet been demonstrated in *Aquilegia*, chromosome four is enriched for the defense

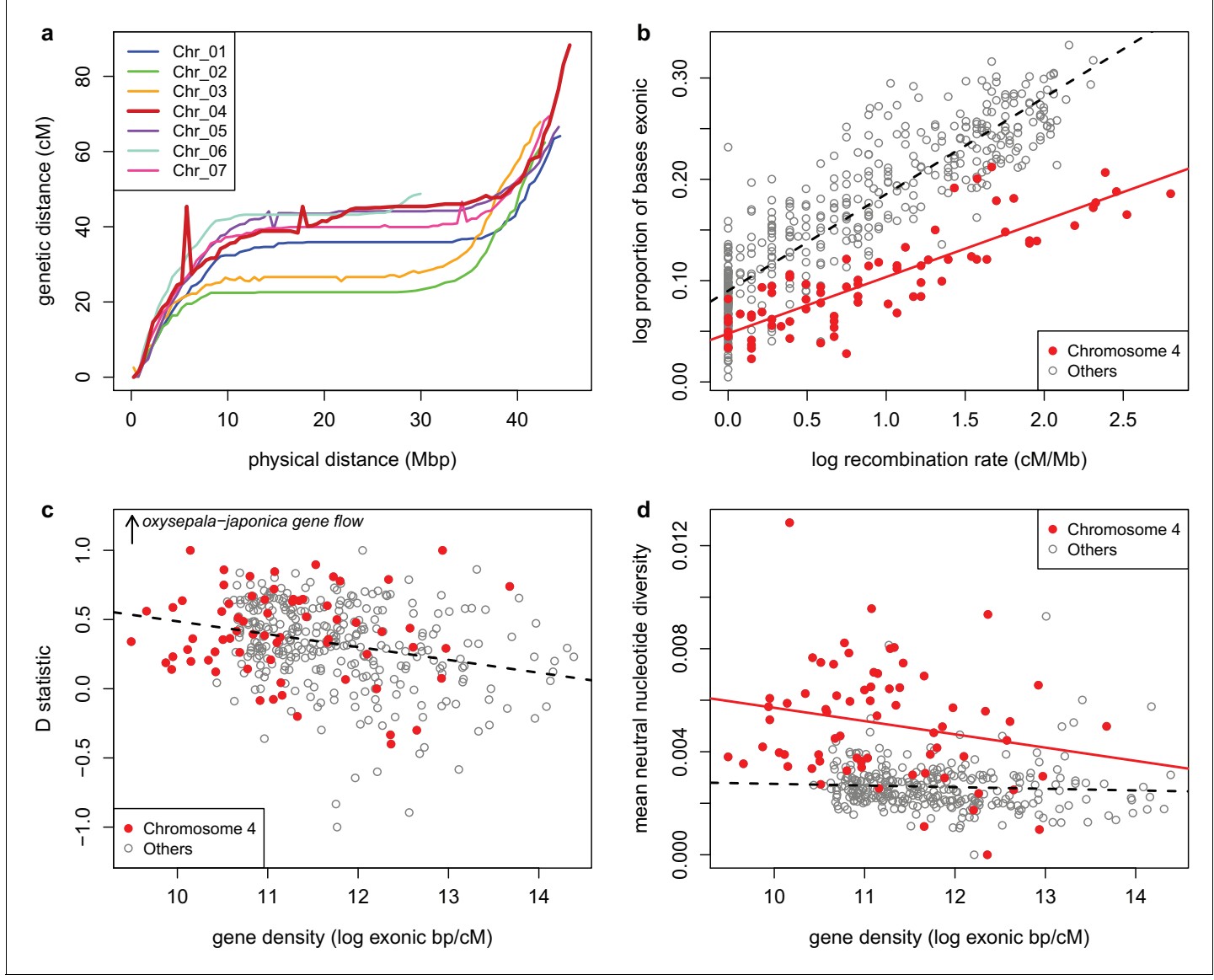

**Figure 7.** Recombination and selection on chromosome four (a) Physical vs. genetic distance for all chromosomes calculated in an *A. formosa* x *A. pubescens* mapping population. High nucleotide diversity on chromosome four was also observed in parental plants of this population (*Figure 7—figure supplement 1*). (b) Relationship between gene density (proportion exonic) and recombination rate (main effect *p*-value < 2 x 10⁻¹⁶, chromosome four effect *p*-value < 2 x 10⁻¹⁶, interaction *p*-value < 1.936 x 10⁻¹¹, adjusted $R^2$ = 0.8045). (c) Relationship between gene density and D statistic for *A. oxysepala* and *A. japonica* gene flow. (d) Relationship between gene density and mean neutral nucleotide diversity. *Figure 7—source data 1*.

DOI: https://doi.org/10.7554/eLife.36426.018

The following source data and figure supplements are available for figure 7:

**Source data 1.** (Physical and genetic distance for *A.formosa x A.pubescens* markers).
DOI: https://doi.org/10.7554/eLife.36426.021
**Figure supplement 1.** Polymorphism in the *A. formosa x A. pubescens* mapping population.
DOI: https://doi.org/10.7554/eLife.36426.019
**Figure supplement 2.** Distribution of gene expression values by chromosome.
DOI: https://doi.org/10.7554/eLife.36426.020

gene GO category, which encompasses *R* genes (*Table 1*). However, while significant, this enrichment involves a relatively small number of genes (less than 2% of genes on chromosome four) and is therefore unlikely to completely explain the polymorphism pattern (*Nordborg and Innan, 2003*).

**Table 1.** GO term enrichment on chromosome four

| GO | Corrected P-value | Number on Chr_04 Observed | Number on Chr_04 Expected | Percent of Chr_04 genes | GO term |
|---|---|---|---|---|---|
| 0043531 | $5.61 \times 10^{-79}$ | 140 | 9 | 7.57 | ADP binding |
| 0016705 | $4.40 \times 10^{-48}$ | 179 | 39 | 9.68 | Oxidoreductase activity, acting on paired donors, with incorporation or reduction of molecular oxygen |
| 0004497 | $7.19 \times 10^{-46}$ | 158 | 32 | 8.55 | Monooxygenase activity |
| 0005506 | $2.73 \times 10^{-41}$ | 181 | 46 | 9.79 | Iron ion binding |
| 0020037 | $2.57 \times 10^{-37}$ | 186 | 53 | 10.06 | Heme binding |
| 0010333 | $1.72 \times 10^{-15}$ | 39 | 4 | 2.11 | Terpene synthase activity |
| 0016829 | $2.08 \times 10^{-13}$ | 39 | 5 | 2.11 | Lyase activity |
| 0055114 | $9.53 \times 10^{-10}$ | 247 | 149 | 13.36 | Oxidation-reduction process |
| 0016747 | $6.66 \times 10^{-5}$ | 44 | 16 | 2.38 | Transferase activity, transferring acyl groups other than amino-acyl groups |
| 0000287 | $1.23 \times 10^{-4}$ | 42 | 15 | 2.27 | Magnesium ion binding |
| 0008152 | $2.56 \times 10^{-4}$ | 137 | 83 | 7.41 | Metabolic process |
| 0006952 | $3.60 \times 10^{-4}$ | 32 | 10 | 1.73 | Defense response |
| 0004674 | $4.52 \times 10^{-4}$ | 23 | 5 | 1.24 | Protein serine/threonine kinase activity |
| 0016758 | $1.35 \times 10^{-3}$ | 44 | 18 | 2.38 | Transferase activity, transferring hexosyl groups |
| 0005622 | $4.14 \times 10^{-3}$ | 14 | 42 | 0.76 | Intracellular |
| 0008146 | $2.68 \times 10^{-2}$ | 9 | 1 | 0.49 | Sulfotransferase activity |
| 0016760 | $3.72 \times 10^{-2}$ | 12 | 2 | 0.65 | Cellulose synthase (UDP-forming) activity |

DOI: https://doi.org/10.7554/eLife.36426.022

Another potential explanation is reduced purifying selection. In fact, several characteristics of chromosome four suggest that it could experience less purifying selection than the rest of the genome. Gene density is markedly lower (*Table 2* and *Figure 7b*), it harbors a higher proportion of repetitive sites (*Table 2*), and is enriched for many transposon families, including Copia and Gypsy elements (*Supplementary file 8*). Additionally, a higher proportion of genes on chromosome four were either not expressed or expressed at a low level (*Figure 7—figure supplement 2*). Gene models on the chromosome were also more likely to contain variants that could disrupt protein function (*Table 2*). Taken together, these observations suggest less purifying selection on chromosome four.

**Table 2.** Content of the *A. coerulea* v3.1 reference by chromosome

| | Chromosome 1 | 2 | 3 | 4 | 5 | 6 | 7 | Genome |
|---|---|---|---|---|---|---|---|---|
| Number of genes | 5041 | 4390 | 4449 | 3149 | 4786 | 3292 | 4443 | 29550 |
| Genes per Mb | 112 | 102 | 104 | 69 | 107 | 108 | 102 | 100 |
| Mean gene length (bp) | 3629 | 3641 | 3689 | 3020 | 3712 | 3620 | 3708 | 3580 |
| Percent repetitive | 38.9 | 41.1 | 39.1 | 54.2 | 39.4 | 39.3 | 40.6 | 42.0 |
| Percent genes with HIGH effect variant | 25.3 | 23.8 | 23.6 | 32.3 | 24.1 | 22.1 | 23.6 | 24.7 |
| Percent GC | 36.8 | 37.0 | 36.9 | 37.0 | 37.1 | 36.8 | 36.8 | 37.0 |

DOI: https://doi.org/10.7554/eLife.36426.023

Reduced purifying selection could also explain the putatively higher gene flow between *A. oxysepala* and *A. japonica* on chromosome four (*Figure 6*); the chromosome would be more permeable to gene flow if loci involved in the adaptive radiation were preferentially located on other chromosomes. Indeed, focusing on *A. oxysepala/A. japonica* gene flow, we found a negative relationship between introgression and gene density in the *Aquilegia* genome (*Figure 7c*, *p*-value = $2.202 \times 10^{-7}$, adjusted R-squared = 0.068), as would be expected if purifying selection limited introgression. Notably, this relationship is the same for chromosome four and the rest of the genome (*p*-value = 0.051), suggesting that gene flow on chromosome four is higher simply because the gene density is lower.

However, the picture is very different for nucleotide diversity. While there is a negative relationship between gene density and neutral nucleotide diversity genome-wide (*p*-value = $5.174 \times 10^{-6}$, adjusted $R^2$ = 0.052), more careful analysis reveals that chromosome four has a completely different distribution from the rest of the genome (*Figure 7d*, *p*-value < $2 \times 10^{-16}$). In both cases, there is a weak (statistically insignificant) negative relationship between gene density and nucleotide diversity (chromsome four: *p*-value = 0.0814, adjusted $R^2$ = 0.0303, rest of the genome: *p*-value = 0.315 , adjusted $R^2$ = $3.373 \times 10^{-5}$), but nucleotide diversity is consistently much higher for chromosome four. Thus the genome-wide relationship reflects this systematic difference between chromosome four and the rest of the genome, and gene density differences alone are insufficient to explain higher polymorphism on chromosome four. Therefore, if reduced background selection explains higher polymorphism on this chromosome, something other than gene density must distinguish it from the rest of the genome. As noted above, there is reason to believe that purifying selection, in general, is lower on this chromosome.

For comparison with data from other organisms, we performed the partial correlation analysis of *Corbett-Detig et al. (2015)*. Here we found a significant relationship between neutral diversity and recombination rate (without chromosome four, Kendall's tau = 0.222, *p*-value = $3.804 \times 10^{-6}$), putting *Aquilegia* on the higher end of estimates of the strength of linked selection in herbaceous plants.

While selection during the *Aquilegia* radiation contributes to the pattern of polymorphism on chromosome four, the pattern itself predates the radiation. Divergence between *Aquilegia* and *Semiaquilegia* is higher on chromosome four (2.77% on chromosome four, 2.48% genome-wide, *Table 3*), as is nucleotide diversity within *Semiaquilegia* (0.16% chromosome four, 0.08% genome-wide, *Table 3*). This suggests that the variant evolutionary history of chromosome four began before the *Aquilegia/Semiaquilegia* split.

## The 35S and 5S rDNA loci are uniquely localized to chromosome four

The observation that one *Aquilegia* chromosome is different from the others is not novel; previous cytological work described a single nucleolar chromosome that appeared to be highly heterochromatic (*Linnert, 1961*). Using fluorescence in situ hybridization (FISH) with rDNA and chromosome four-specific bulked oligo probes (*Han et al., 2015*), we confirmed that both the 35S and 5S rDNA loci were localized uniquely to chromosome four in two *Aquilegia* species and *S. adoxoides* (*Figure 8*). The chromosome contained a single large 35S repeat cluster proximal to the centromeric region in all three species. Interestingly, the 35S locus in *A. formosa* was larger than that of the other two species and formed variable bubbles and fold-backs on extended pachytene chromosomes similar to structures previously observed in *Aquilegia* hybrids (*Linnert, 1961*) (*Figure 8*, last panels). The 5S rDNA locus was also proximal to the centromere on chromosome four, although slight differences in the number and position of the 5S repeats between species highlight the dynamic nature of this gene cluster. However, no chromosome appeared to be more heterochromatic than others in our

**Table 3.** Population genetics parameters for *Semiaquilegia* by chromosome

| | Percent pairwise differences | | | | | | | |
| | Chromosome | | | | | | | |
| | 1 | 2 | 3 | 4 | 5 | 6 | 7 | Genome |
|---|---|---|---|---|---|---|---|---|
| Polymorphism within *Semiaquilegia* | 0.079 | 0.085 | 0.081 | 0.162 | 0.076 | 0.078 | 0.071 | 0.082 |
| Divergence between *Aquilegia* and *Semiaquilegia* | 2.46 | 2.47 | 2.47 | 2.77 | 2.48 | 2.47 | 2.47 | 2.48 |

DOI: https://doi.org/10.7554/eLife.36426.024

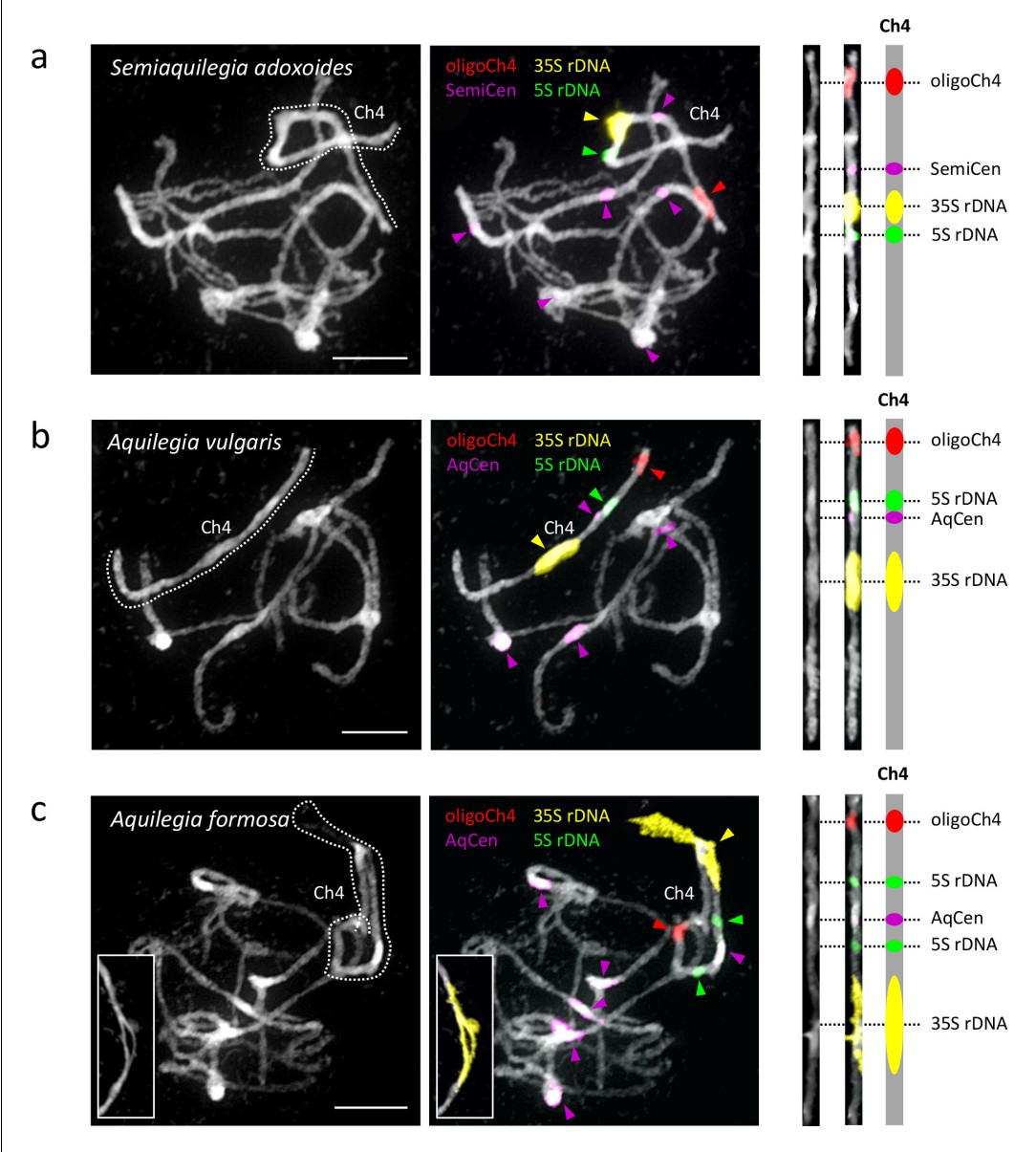

**Figure 8.** Cytogenetic characterization of chromosome four in *Semiaquilegia* and *Aquilegia* species. Pachytene chromosome spreads were probed with probes corresponding to oligoCh4 (red), 35S rDNA (yellow), 5S rDNA (green) and two (peri)centromeric tandem repeats (pink). Chromosomes were counterstained with DAPI. Scale bars = 10 μm.

DOI: https://doi.org/10.7554/eLife.36426.025

The following figure supplement is available for figure 8:

**Figure supplement 1.** Immunodetection of anti-5mC antibody.

DOI: https://doi.org/10.7554/eLife.36426.026

analyses (*Figure 8*); FISH with 5-methylcytosine antibody showed no evidence for hypermethylation on chromosome four (*Figure 8—figure supplement 1*) and GC content was similar for all chromosomes (*Table 2*). However, similarities in chromosome four organization across all three species reinforce the idea that the exceptionality of this chromosome predated the *Aquilegia/Semiaquilegia* split and raise the possibility that rDNA clusters could have played a role in the variant evolutionary history of chromosome four.

## Discussion

We constructed a reference genome for the horticultural cultivar *Aquilegia coerulea* 'Goldsmith' and resequenced ten *Aquilegia* species with the goal of understanding the genomics of ecological speciation in this rapidly diversifying lineage. Although our reference genome size is smaller than previous estimates (~300 Mb versus ~500 Mb, [*Bennett et al., 1982*; *Bennett and Leitch, 2011*]), the completeness and accuracy of our assembly (*Supplementary file 4*), as well as consistency between reference and *k*-mer based estimates of genome size (*Supplementary file 9*), suggest that this difference is likely due to highly repetitive content, including the large rDNA loci on chromosome four.

Variant sharing across the *Aquilegia* genus is widespread and deep, even across exceptionally large geographical distances. Although much of this sharing is presumably due to stochastic processes, as expected given the rapid time-scale of speciation, asymmetry of allele sharing demonstrates that the process of speciation has been reticulate throughout the genus, and that gene flow has been a common feature. *Aquilegia* species diversity therefore appears to be an example of ecological speciation, rather than being driven by the development of intrinsic barriers to gene flow (*Coyne et al., 2004*; *Schluter and Conte, 2009*; *Seehausen et al., 2014*). In the future, studies incorporating more taxa and/or population-level variation will provide additional insight into the dynamics of this process. Given the extent of variant sharing, it will be also be interesting to explore the role of standing variation and admixture in adaptation throughout the genus.

Our analysis also led to the remarkable discovery that the evolutionary history of an entire chromosome differed from that of the rest of the genome. The average level of polymorphism on chromosome four is roughly twice that of the rest of the genome and gene trees on this chromosome appear to reflect a different species relationship (*Figure 3*). To the best of our knowledge, with the possible exception of sex chromosomes (*Toups and Hahn, 2010*; *Nam et al., 2015*), such chromosome-wide patterns have never been observed before (although recombination has been shown to affect hybridization; see *Schumer et al., 2018*). Importantly, this chromosome is large and appears to be freely recombining, implying that these differences are unlikely to be due to a single evolutionary event, but rather reflect the accumulated effects of evolutionary forces acting differentially on the chromosome.

While no single explanation for the elevated polymorphism on chromosome four has emerged, selection clearly plays a role. Our results demonstrate that chromosome four could be affected by balancing selection as well as by reduced purifying and/or background selection. Future work will focus on clarifying the role and importance of each of these types of selection, and determining whether the rapid adaptive radiation in *Aquilegia* has played a role in accelerating the differences between chromosome four and the rest of the genome.

The chromosome four patterns, appear to predate the *Aquilegia* adaptive radiation, however, extending at least into the genus *Semiaquilegia*. Differences in gene content may thus be a proximal explanation for the higher polymorphism levels on chromosome four, but we still lack an explanation for why these differences would have been established on chromosome four in the first place. One possibility is that chromosome four is a reverted sex chromosome, a phenomenon that has been observed in *Drosophila* (*Vicoso and Bachtrog, 2013*). Although species with separate sexes exist in the Ranunculaceae, these transitions seem to be recent (*Soza et al., 2012*), and all *Aquilegia* and *Semiaquilegia* species are hermaphroditic. Furthermore, no heteromorphic sex chromosomes have been observed in the Ranunculales (*Westergaard, 1958*; *Ming et al., 2011*), making this an unlikely hypothesis. It has also been suggested that chromosome four is a fusion of two homeologous chromosomes (*Linnert, 1961*), as could result from the ancestral whole genome duplication (*Cui et al., 2006*; *Vanneste et al., 2014*; *Tiley et al., 2016*), however, analysis of synteny blocks shows that this is not the case (*Aköz and Nordborg, 2018*).

B chromosomes also have evolutionary histories that differ from those of other chromosomes. Like chromosome four, B chromosomes accumulate repetitive sequences and frequently contain rDNA loci (*Jones, 1995*; *Green, 1990*; *Valente et al., 2017*). However, chromosome four does not appear to be supernumerary, and unlike B chromosomes which seem to have only a few loci, chromosome four contains thousands of coding sequences (*Table 2*). Again, while it is impossible to rule out the hypothesis that chromosome four has been impacted by the reincorporation of B chromosomes into the A genome, this would be a novel phenomenon.

It is tempting to speculate that the distinct evolutionary history of chromosome four is connected to its large rDNA repeat clusters. Although rDNA clusters in *Aquilegia* and *Semiaquilegia* are consistently found on chromosome four, cytology demonstrates that the exact location of these loci is dynamic. Could the movement of these components somehow contribute to an accumulation of structural variants, copy number variants, and repeats that make chromosome four an inhospitable and unreliable place to harbor critical coding sequences? If so, then forces of genome evolution could underlie the more proximal causes (lower gene content and reduced selection) of increased polymorphism on chromosome four.

rDNA clusters could also have played a role in initiating chromosome four's different evolutionary history. Cytological (*Langlet, 1927*; *Langlet, 1932*) and phylogenetic (*Ro et al., 1997*; *Wang et al., 2009*; *Cossard et al., 2016*) work separates the Ranunculaceae into two main subfamilies marked by different base chromosome numbers: the Thalictroideae (T-type, base n = 7, including *Aquilegia* and *Semiaquilegia*) and the Ranunculoideae (R-type, predominantly base n = 8). In the three T-type species tested here, the 35S is proximal to the centromere, a localization seen for only 3.5% of 35S sites reported in higher plants (*Roa and Guerra, 2012*). In contrast, all R-type species examined have terminal or subterminal 45S loci (*Hizume et al., 2013*; *Mlinarec et al., 2006*; *Weiss-Schneeweiss et al., 2007*; *Liao et al., 2008*). Given that 35S repeats can be fragile sites (*Huang et al., 2008*) and 35S rDNA clusters and rearrangement breakpoints co-localize (*Cazaux et al., 2011*), a 35S-mediated chromosomal break could explain differences in base chromosome number between R-type and T-type species. If the variant history of chromosome four can be linked to this this R- vs T-type split, this could implicate chromosome evolution as the initiator of chromosome four's variant history. Comparative genomics work within the Ranunculaceae will therefore be useful for understanding the role that rDNA repeats have played in chromosome evolution and could provide additional insight into how rDNA could have contributed to chromosome four's variant evolutionary history.

In conclusion, the *Aquilegia* genus is a beautiful example of adaptive radiation through ecological speciation. Although our current genome analyses based on a limited number of individuals and species, we see evidence that the radiation was shaped by introgression, selection, and the presence of abundant standing variation. On-going work focuses on understanding the contributions of each of these factors to adaptation in *Aquilegia* using population and quantitative genetics. Additionally, the unexpected variant evolutionary history of chromosome four, while still a mystery, illustrates that standard population genetics models are not always sufficient to the explain the pattern of variation across the genome. Future studies of chromosome four have the potential to increase our understanding of how genome evolution, chromosome evolution, and population genetics interact to generate organismal diversity.

## Materials and methods

### Genome sequencing, assembly, and annotation

#### Sequencing

Sequencing was performed on *Aquilegia coerulea* cv 'Goldsmith', an inbred line constructed and provided by Todd Perkins of Goldsmith Seeds (now part of Syngenta). The line was of hybrid origin of multiple taxa and varieties of *Aquilegia* and then inbred. The sequencing reads were collected with standard Sanger sequencing protocols at the Department of Energy Joint Genome Institute in Walnut Creek, California and the HudsonAlpha Institute for Biotechnology. Libraries included two 2.5 Kb libraries (3.36x), two 6.5 Kb libraries (3.70x), two 33 Kb insert size fosmid libraries (0.36x), and one 124 kb insert size BAC library (0.17x). The final read set consists of 4,773,210 reads for a total of 2.988 Gb high quality bases (*Supplementary file 1*).

#### Genome assembly and construction of pseudomolecule chromosomes

Sequence reads (7.59x assembled sequence coverage) were assembled using our modified version of Arachne v.20071016 (*Jaffe et al., 2003*) with parameters maxcliq1 = 120 n_haplotypes = 2 max_bad_look = 2000 START = SquashOverlaps BINGE_AND_PURGE_2HAP = True.

This produced 2529 scaffolds (10,316 contigs), with a scaffold N50 of 3.1 Mb, 168 scaffolds larger than 100 kb, and total genome size of 298.6 Mb (*Supplementary file 2*). Two genetic maps (*A.*

coerulea 'Goldsmith' x *A. chrysantha* and *A. formosa* x *A. pubescens*) were used to identify 98 mis-joins in the initial assembly. Misjoins were identified by a linkage group/syntenic discontinuity coincident with an area of low BAC/fosmid coverage. A total of 286 scaffolds were ordered and oriented with 279 joins to form seven chromosomes. Each chromosome join is padded with 10,000 Ns. The remaining scaffolds were screened against bacterial proteins, organelle sequences, GenBank nr and removed if found to be a contaminant. Additional scaffolds were removed if they (a) consisted of >95% 24mers that occurred four other times in scaffolds larger than 50 kb (957 scaffolds, 6.7 Mb), (b) contained only unanchored RNA sequences (14 scaffolds, 651.9 Kb), or (c) were less than 1 kb in length (303 scaffolds). Significant telomeric sequence was identified using the TTTAGGG repeat, and care was taken to make sure that it was properly oriented in the production assembly. The final release assembly (*A. coerulea* 'Goldsmith' v3.0) contains 1034 scaffolds (7930 contigs) that cover 291.7 Mb of the genome with a contig N50 of 110.9 kb and a scaffold L50 of 43.6 Mb (*Supplementary file 3*).

## Validation of genome assembly

Completeness of the euchromatic portion of the genome assembly was assessed using 81,617 full length cDNAs (*Kramer and Hodges, 2010*). The aim of this analysis is to obtain a measure of completeness of the assembly, rather than a comprehensive examination of gene space. The cDNAs were aligned to the assembly using BLAT (*Kent, 2002*) (Parameters: -t = dna –q = rna –extendThroughN -noHead) and alignments >=90% base pair identity and >=85% EST coverage were retained. The screened alignments indicate that 79,626 (98.69%) of the full length cDNAs aligned to the assembly. The cDNAs that failed to align were checked against the NCBI nucleotide repository (nr), and a large fraction were found to be arthropods (*Acyrthosiphon pisum*) and prokaryotes (*Acidovorax*).

A set of 23 BAC clones were sequenced in order to assess the accuracy of the assembly. Minor variants were detected in the comparison of the fosmid clones and the assembly. In all 23 BAC clones, the alignments were of high quality (<0.35% bp error), with an overall bp error rate (including marked gap bases) in the BAC clones of 0.24% (1,831 discrepant bp out of 3,063,805; *Supplementary file 4*).

## Genomic repeat and transposable element prediction

Consensus repeat families were predicted de novo for the *A. coerulea* v3.0 assembly by the RepeatModeler pipeline (*Smit and Hubley, 2015*). These consensus sequences were annotated for PFAM and Panther domains, and any sequences known to be associated with non-TE function were removed. The final curated library was used to generate a softmasked version of the *A. coerulea* 'Goldsmith' v3.0 assembly.

## Transcript assembly and gene model annotation

A total of 246 million paired-end and a combined 1 billion single-end RNAseq reads from a diverse set of tissues and related *Aquilegia* species (*Supplementary file 5* and BioProject ID PRJNA270946 [*Yant et al., 2015*]) were assembled using PERTRAN (*Shu et al., 2013*) to generate a candidate set containing 188,971 putative transcript assemblies. The PERTRAN candidate set was combined with 115,000 full length ESTs (the 85,000 sequence cDNA library derived from an *A. formosa X A. pubescens* cross (*Kramer and Hodges, 2010*) and 30,000 Sanger sequences of *A. formosa* sequenced at JGI) and aligned against the v3.0 assembly of the *A. coerulea* 'Goldsmith' genome by PASA (*Haas et al., 2003*).

Loci were determined by BLAT alignments of above transcript assemblies and/or BLASTX of the proteomes of a diverse set of angiosperms (*Arabidopsis thaliana* TAIR10, *Oryza sativa* v7, *Glycine max* Wm82.a2.v1, *Mimulus guttatus* v2, *Vitus vinifera* Genoscape.12X and *Poplar trichocarpa* v3.0). These protein homology seeds were further extended by the EXONERATE algorithm. Gene models were predicted by homology-based predictors, FGENESH+ (*Salamov and Solovyev, 2000*), FGENESH_EST (similar to FGENESH+, but using EST sequence to model splice sites and introns instead of putative translated sequence), and GenomeScan (*Yeh et al., 2001*).

The final gene set was selected from all predictions at a given locus based on evidence for EST support or protein homology support according to several metrics, including Cscore, a protein

BLASTP score ratio to homology seed mutual best hit (MBH) BLASTP score, and protein coverage, counted as the highest percentage of protein model aligned to the best of its angiosperm homologs. A gene model was selected if its Cscore was at least 0.40 combined with protein homology coverage of at least 45%, or if the model had EST coverage of at least 50%. The predicted gene set was also filtered to remove gene models overlapping more than 20% with a masked RepeatModeler consensus repeat region of the genome assembly, except for such cases that met more stringent score and coverage thresholds of 0.80% and 70% respectively. A final round of filtering to remove putative transposable elements was conducted using known TE PFAM and Panther domain homology present in more than 30% of the length of a given gene model. Finally, the selected gene models were improved by a second round of the PASA algorithm, which potentially included correction to selected intron splice sites, addition of UTR, and modeling of alternative spliceforms.

The resulting annotation and the *A. coerulea* 'Goldsmith' v3.0 assembly make up the *A. coerulea* 'Goldsmith' v3.1 genome release, available on Phytozome (https://phytozome.jgi.doe.gov/)

## Sequencing of species individuals

### Sequencing, mapping and variant calling

Individuals of 10 *Aquilegia* species and *Semiaquilegia adoxoides* were resequenced (*Figure 1* and *Figure 1—figure supplement 1*). One sample (*A. pubescens*) was sequenced at the Vienna Biocenter Core Facilities Next Generation Sequencing (NGS) unit in Vienna, Austria and the others were sequenced at the Department of Energy Joint Genome Institute (Walnut Creek, CA, USA). All libraries were prepared using standard/slightly modified Illumina protocols and sequenced using paired-end Illumina sequencing. *Aquilegia* species read length was 100 bp, the *S. adoxoides* read length was 150 bp, and samples were sequenced to a depth of 58-124x coverage (*Supplementary file 10*). Sequences were aligned against *A. coerulea* 'Goldsmith' v3.1 with bwa mem (bwa mem -t 8 p -M) (*Li and Durbin, 2009*; *Li, 2013*). Duplicates and unmapped reads were removed with SAMtools (*Li et al., 2009*). Picardtools (*Picard Tools, 2018*) was used to clean the resulting bam files (CleanSam.jar), to remove duplicates (MarkDuplicates.jar), and to fix mate pair problems (FixMateInformation.jar). GATK 3.4 (*McKenna et al., 2010*; *DePristo et al., 2011*) was used to identify problem intervals and do local realignments (RealignTargetCreator and IndelRealigner). The GATK Haplotype Caller was used to generate gVCF files for each sample. Individual gVCF files were merged and GenotypeGVCFs in GATK was used to call variants.

### Variant filtration

Variants were filtered to identify positions in the single-copy genome that could be reliable called across all *Aquilegia* individuals. Variant Filtration in GATK 3.4 (*McKenna et al., 2010*; *DePristo et al., 2011*) was used to filter multialleleic sites, indels ± 10 bp, sites identified with RepeatMasker (*Smit et al., 2015*), and sites in previously-determined repetitive elements (see *'Genomic repeat and transposable element prediction' above*). We required a minimum coverage of 15 in all samples and a genotype call (either variant or non-variant) in all accessions. Sites with less than -0.5x log median coverage or greater than 0.5x log median coverage in any sample were also removed. A table of the number of sites removed by each filter is in *Supplementary file 11*.

### Polarization

*S. adoxoides* was added to the *Aquilegia* individual species data set and the above filtration was repeated (*Supplementary file 12*). The resulting variants were then polarized against *S. adoxoides*, resulting in nearly 1.5 million polarizable variant positions. A similar number of derived variants was detected in all species (*Supplementary file 13*), suggesting no reference bias resulting from the largely North American provenance of the *A. coerulea* v3.1reference sequence used for mapping.

## Evolutionary analysis

### Basic population genetics

Basic population genetics parameters including nucleotide diversity (polymorphism and divergence) and $F_{ST}$ were calculated using custom scripts in R (*R Core Team, 2014*). Nucleotide diversity was calculated as the percentage of pairwise differences in the mappable part of the genome. $F_{ST}$ was calculated as in *Hudson et al. (1992)*. To identify fourfold degenerate sites, four pseudo-vcfs

replacing all positions with A,T,C, or G, respectively, were used as input into SNPeff (*Cingolani et al., 2012*) to assess the effect of each pseudo-variant in reference to the *A. coerulea* 'Goldsmith' v3.1 annotation. Results from all four output files were compared to identify genic sites that caused no predicted protein changes.

## Tree and cloudogram construction

Trees were constructed using a concatenated data set of all nonfiltered sites, either genome-wide or by chromosome. Neighbor joining (NJ) trees were made using the ape (*Paradis et al., 2004*) and phangorn (*Schliep, 2011*) packages in R (*R Core Team, 2014*) using a Hamming distance matrix and the nj command. RAxML trees were constructed using the default settings in RAxML (*Stamata-kis, 2014*). All trees were bootstrapped 100 times. The cloudogram was made by constructing NJ trees using concatenated nonfiltered SNPs in non-overlapping 100 kb windows across the genome (minimum of 100 variant positions per window, 2387 trees total) and plotted using the densiTree package in phangorn (*Schliep, 2011*).

## Differences in subtree frequency by chromosome

For each of the 217 subtrees that had been observed in the cloudogram, we calculated the proportion of window trees on each chromosome containing the subtree of interest and performed a test of equal proportions (prop.test in R [*R Core Team, 2014*]) to determine whether the prevalence of the subtree varied by chromosome. For significantly-varying subtrees, we then performed another test of equal proportions (prop.test in R [*R Core Team, 2014*]) to ask whether subtree proportion on each chromosome was different from the genome-wide proportion. The appropriate Bonferroni multiple testing correction was applied to *p*-values obtained in both tests (n = 217 and n = 70, respectively).

## Tests of D statistics

D statistics tests were performed in ANGSD (*Korneliussen et al., 2014*) using non-filtered sites only. ANGSD ABBABABA was run with a block size of 100000 and results were bootstrapped. Tests were repeated using only fourfold degenerate sites.

## Modelling effects of migration rate and effective population size

Using the markovchain (*Spedicato, 2017*) package in R (*R Core Team, 2014*), we simulated a simple coalescent model with the assumptions as follows: (1) population size is constant (N alleles) at all times, (2) *A. oxysepala* split from the population ancestral to *A. sibirica* and *A. japonica* at generation $t_2 = 2*t$, (3) *A. sibirica* and *A. japonica* split from each other at generation $t_1 = t$, and (4) there was gene flow between *A. oxysepala* and *A. japonica* between t = 0 and $t_1$. A first Markov Chain simulated migration with symmetric gene flow ($m_1 = m_2$) and coalescence between t = 0 and $t_1$ (*Five-State Markov Chain*, *Supplementary file 14*). This process was run for T (t*N) generations to get the starting probabilities for the second process, which simulated coalescence between $t_1$ and $t_2 + 1$ (*Eight-State Markov Chain*, *Supplementary file 15*). The second process was run for N generations. After first identifying a combination of parameters that minimized the difference between simulated versus observed gene genealogy proportions, we then reran the process with increased migration rate and/or N to check if simulated proportions matched observed chromosome four-specific proportions. We also ran the initial chain under two additional models of gene flow: unidirectional ($m_2 = 0$) and asymmetric ($m_1 = 2*m_2$).

## Robustness of chromosome four patterns to filtration

Variant filtration as outlined above was repeated with a stringent coverage filter (keeping only positions with ±0.15x log median coverage in all samples) and nucleotide diversity per chromosome was recalculated. Nucleotide diversity per chromosome was also recalculated after removal of copy number variants detected by the readDepth package (*Miller et al., 2011*) in R (*R Core Team, 2014*), after the removal of tandem duplicates as determined by running DAGchainer (*Haas et al., 2004*) on *A. coerulea* 'Goldsmith' v3.1 in CoGe SynMap (*Lyons and Freeling, 2008*), as well as after the removal of heterozygous variants for which both alleles were not supported by an equivalent number of reads (a log read number ratio <−0.3 or >0.3).

## Construction of an *A. formosa* x *A. pubescens* genetic map

Mapping and variant detection

Construction of the *A. formosa* x *A. pubescens* F2 cross was previously described in *Hodges et al. (2002)*. One *A. pubescens* F0 line (pub.2) and one *A. formosa* F0 line (form.2) had been sequenced as part of the species resequencing explained above. Libraries for the other *A. formosa* F0 (form.1) were constructed using a modified Illumina Nextera library preparation protocol (*Baym et al., 2015*) and sequenced at the Vincent J. Coates Genomics Sequencing Laboratory (UC Berkeley). Libraries for the other *A. pubescens* F0 (pub.1), and for both F1 individuals (F1.1 and F1.2), were prepared using a slightly modified Illumina Genomic DNA Sample preparation protocol (*Rabanal et al., 2017*) and sequenced at the Vienna Biocenter Core Facilities Next Generation Sequencing (NGS) unit in Vienna, Austria. All individual libraries were sequenced as 100 bp paired-end reads on the Illumina HiSeq platform to 50-200x coverage. A subset of F2s were sequenced at the Vienna Biocenter Core Facilities Next Generation Sequencing (NGS) unit in Vienna, Austria (70 lines). Libraries for the remaining F2s (246 lines) were prepared and sequenced by the Department of Energy Joint Genome Institute (Walnut Creek, CA). All F2s were prepared using the Illumina multiplexing protocol and sequenced on the Illumina HiSeq platform to generate 100 bp paired end reads. Samples were 96-multiplexed to generate about 1-2x coverage. Sequences for all samples were aligned to the *A. coerulea* 'Goldsmith' v3.1 reference genome using bwamem with default parameters (*Li and Durbin, 2009*; *Li, 2013*). SAMtools 0.1.19 (*Li et al., 2009*) mpileup (-q 60 -C 50 -B) was used to call variable sites in the F1 individuals. Variants were filtered for minimum base quality and minimum and maximum read depth (-Q 30, -d20, -D150) using SAMtools varFilter. Variable sites that had a genotype quality of 99 in the F1s were genotyped in F0 plants to generate a set of diagnostic alleles for each parent of origin. To assess nucleotide diversity, F0 and F1 samples were additionally processed with the mapping and variant calling pipeline as described for species samples above.

Genotyping of F2s, genetic map construction, and recombination rate estimation

F2s were genotyped in genomic bins of 0.5 Mb in regions of moderate to high recombination and 1 Mb in regions with very low or no recombination, as estimated by the *A. coerulea*' Goldsmith' x *A. chrysantha* cross used to assemble the A. coerulea 'Goldsmith' v3.1 reference genome (see '*Genome assembly and construction of pseudomolecule chromosomes*'). Ancestry of each bin was independently determined for each of the four parents. The ratio of reads containing a diagnostic allele/reads potentially containing a diagnostic allele was calculated for each parent in each bin. If this ratio was between 0.4 and 0.6, the bin was assigned to the parent containing the diagnostic allele. Bins with ratios between 0.1 and 0.4 were considered more closely to determine whether they represented a recombination event or discordance between the physical map of *A. formosa*/*A. pubescens* and the *A. coerulea* 'Goldsmith' v3.1 reference genome. If a particular bin had intermediate frequencies for many F2 individuals, indicative of a map discordance, bin margins were adjusted to capture the discordant fragment and allow it to independently map during genetic map construction.

Bin genotypes were used as markers to assemble a genetic map using R/qtl v.1.35–3 (*Broman et al., 2003*). Genetic maps were initially constructed for each chromosome of a homolog pair in the F2s. After the two F1 homologous chromosome maps were estimated, data from each chromosome was combined to estimate the combined genetic map. Several bins in which genotypes could not accurately be determined either due to poor read coverage, lack of diagnostic SNPs, or unclear discordance with the reference genome were dropped from further recombination rate analysis.

To measure recombination in each F1 parent, genetic maps were initially constructed for each chromosome of a homolog pair in the F2s. After the two F1 homologous chromosome maps were estimated, data from each chromosome was combined to estimate the combined genetic map. In order to calculate the recombination rate for each bin, a custom R script (*R Core Team, 2014*) was written to count the number of recombination events in the bin which was then averaged by the number of haploid genomes assessed (n = 648). To calculate cM per Mb, this number was multiplied by 100 and divided by the bin size in Mb.

## Chromosome four gene content and background selection

Unless noted, all analyses were done in R (*R Core Team, 2014*).

### Gene content, repeat content, and variant effects

Gene density and mean gene length were calculated considering primary transcripts only. Percent repetitive sequence was determined from annotation. The effects of variants was determined with SNPeff (*Cingolani et al., 2012*) using the filtered variant data set and primary transcripts.

### Repeat family content

The RepeatClassifier utility from RepeatMasker (*Smit et al., 2015*) was used to assign *A. coerulea* v3.1 repeats to known repetitive element classes. For each of the 38 repeat families identified, the insertion rate per Mb was calculated for each chromosome and a permutation test was performed to determine whether this proportion was significantly different on chromosome four versus genome-wide. Briefly, we ran 1000 simulations to determine the number of insertions expected on chromosome four if insertions occurred randomly at the genome-wide insertion rate and then compared this distribution with the observed copy number in our data.

### GO term enrichment

A two-sided Fisher's exact test was performed for each GO term to test whether the term made up a more extreme proportion on of genes on chromosome four versus the proportion in the rest of the genome. *P*-values were Bonferroni corrected for the number of GO terms (n = 1936).

## Quantification of gene expression

We sequenced whole transcriptomes of sepals from 21 species of *Aquilegia* (*Supplementary file 5*). Tissue was collected at the onset of an thesis and immediately immersed in RNAlater (Ambion) or snap frozen in liquid nitrogen. Total RNA was isolated using RNeasy kits (Qiagen) and mRNA was separated using poly-A pulldown (Illumina). Obtaining amounts of mRNA sufficient for preparation of sequencing libraries required pooling multiple sepals together into a single sample; we used tissue from a single individual when available, but often had to pool sepals from separate individuals into a single sample. We prepared sequencing libraries according to manufacturer's protocols except that some libraries were prepared using half-volume reactions (Illumina RNA-sequencing for *A. coerulea*, and half-volume Illumina TruSeq RNA for all other species). Libraries for *A. coerulea* were sequenced one sample per lane on an Illumina GAII (University of California, Davis Genome Center). Libraries for all other species were sequenced on an Illumina HiSeq at the Vincent J. Coates Genomics Sequencing Laboratory (UC Berkeley), with samples multiplexed using TruSeq indexed adapters (Illumina). Reads were aligned to *A. coerulea* 'Goldsmith' v3.1 using bwa aln and bwa samse (*Li and Durbin, 2009*). We processed alignments with SAMtools (*Li and Durbin, 2009*) and custom scripts were used to count the number of sequence reads per transcript for each sample. Reads that aligned ambiguously were probabilistically assigned to a single transcript. Read counts were normalized using calcNormFactors and cpm functions in the R package edgeR (*Robinson et al., 2010*; *McCarthy et al., 2012*). Mean abundance was calculated for each transcript by first averaging samples within a species, and then averaging across all species.

## Relationships between recombination, gene density, introgression, and neutral nucleotide diversity

Recombination rates were taken from the *A. formosa* x *A. pubescens* analysis described in '*Construction of an A. formosa x A. pubescens genetic map*'. We calculated neutral nucleotide diversity, proportion exonic, and the D statistic for each genomic bin used in constructing the genetic map. Polymorphism at fourfold degenerate sites was calculated per species/individual, and the mean of all 10 species was taken. The number of exonic bases was determined using primary gene models from the *A. coerulea* v3.1 'Goldsmith' annotation and both proportion exonic bases and gene density (exonic bases/cM) were calculated. To obtain window D statistics, we parsed the ANGSD raw output file (*out.abbababa) generated for the test of *A. oxysepala* as the outgroup and *A. japonica* and *A. sibirica* as sister species (described in Tests of D-statistics).

We performed linear models in R (**R Core Team, 2014**), log transforming as appropriate, to look at the relationship between (1) exonic proportion and recombination rate, (2) D statistic and gene density (exonic base pairs per cM), and (3) neutral nucleotide diversity and gene density. For each relationship, we tested two models: one considering the genome as a whole and a second that tested for differences between chromosome four and the rest of the genome, including an interaction term when significant. If the second model showed a significant *p*-value difference for chromosome four, we also performed individual linear models for both chromosome four and the rest of the genome.

We also calculated the partial correlation coefficient between recombination rate and neutral nucleotide diversity accounting for gene density as in *Corbett-Detig et al. (2015)*.

## Genome size determination

Genome size was assessed using the *k*-mer counting method. Histograms of 20-base-pair *k*-mer counts were generated in Jellyfish 2.2.6 (**Marçais and Kingsford, 2011**) using the fastq files of all 10 resequenced *Aquilegia* species. Using these histograms, estimates of genome size and repetitive proportion were made using the findGSE library (**Sun et al., 2018**) in R (**R Core Team, 2014**).

## Cytology

### Chromosome preparation

Inflorescences of the analyzed accessions were fixed in ethanol:acetic acid (3:1) overnight and stored in 70% ethanol at $-20°C$. Selected flower buds were rinsed in distilled water and citrate buffer (10 mM sodium citrate, pH 4.8; $2 \times 5$ min) and incubated in an enzyme mix (0.3% cellulase, cytohelicase, and pectolyase; all Sigma-Aldrich) in citrate buffer at 37°C for 3 to 6 hr. Individual anthers were disintegrated on a microscope slide in a drop of citrate buffer and 15 to 30 µl of 60% acetic acid. The suspension was spread on a hot plate at 50°C for 0.5 to 2 min. Chromosomes were fixed by adding 100 µl of ethanol:acetic acid (3:1). The slide was dried with a hair dryer, postfixed in 4% formaldehyde dissolved in distilled water for 10 min, and air-dried. Chromosome preparations were treated with 100 µg/ml RNase in $2 \times$ sodium saline citrate (SSC; 20x SSC: 3 M sodium chloride, 300 mM trisodium citrate, pH 7.0) for 60 min and with 0.1 mg/ml pepsin in 0.01 M HCl at 37° C for 2 to 5 min; then postfixed in 4% formaldehyde in 2x SSC, and dehydrated in an ethanol series (70%, 90%, and 100%, 2 min each).

### Probe preparation from oligo library

An oligonucleotide library consisting of 20,628 oligonucleotide probes to the 2 Mb region spanning the positions 42–44 Mbp of chromosome four (oligoCh4) was designed and synthesized by MYcroarray (Ann Arbor, MI). This library was used to prepare the chromosome four-specific painting probe (**Han et al., 2015**). Briefly, oligonucleotides were multiplied in two independent amplification steps. First, 0.2 ng DNA from the immortal library was amplified from originally ligated adaptors in emulsion PCR using primers provided by MYcroarray together with the library. Emulsion PCR was used to increase the representativeness of amplified products (**Murgha et al., 2014**). Droplets were generated manually by stirring of oil phase at $1000 \times$ g at 4°C for 10 min and then the aqueous phase was added. 500 ng of amplified product was used as a template for T7 in vitro transcription with MEGAshortscript T7 Kit (Invitrogen) – the second amplification step. RNA was purified on RNeasy spin columns (Qiagen) and labeled in reverse transcription with biotin-labeled R primer. The product – RNA: DNA hybrid – was washed using Zymo Quick-RNA MiniPrep (Zymo Research), hydrolysed by RNase and obtained DNA was cleaned again with Zymo Kit to get the final single-stranded DNA probe.

### Fluorescence in situ hybridization (FISH)

Species resequencing data was used to determing the (peri)centromeric satellite repeats of *Semiaquilegia* (SemiCen) and *Aquilegia* (AqCen) (**Melters et al., 2013**); the detected AqCen sequence corresponds to the previously-described centromeric repeat (**Melters et al., 2013**). Bulked oligonucleotides specific for chromosome four (oligoCh4), (peri)centromeric satellite repeats, *Arabidopsis thaliana* BAC clone T15P10 (AF167571) containing 35S rRNA genes, and *A. thaliana* clone pCT4.2 (M65137) corresponding to a 500 bp 5S rRNA repeat were used as probes. All DNA probes were labeled with biotin-dUTP, digoxigenin-dUTP or Cy3-dUTP by nick translation (*Mandáková and*

*Lysak, 2016*). Selected labeled DNA probes were pooled together, ethanol precipitated, dissolved in 20 µl of 50% formamide, 10% dextran sulfate in 2x SSC and pipetted onto microscope slides. The slides were heated at 80°C for 2 min and incubated at 37°C overnight. Posthybridization washing was performed in 20% formamidein 2x SSC at 42°C (2 × 5 min). Hybridized probes were visualized through fluorescently labeled antibodies against biotin or digoxigenin (*Mandáková and Lysak, 2016*). Chromosomes were counterstained with 4',6-diamidino-2-phenylindole (DAPI, 2 µg/ml) in Vectashield antifade. Fluorescence signals were analyzed and photographed using a Zeiss Axioimager epifluorescence microscope and a CoolCube camera (MetaSystems). Individual images were merged and processed using Photoshop CS software (Adobe Systems). Pachytene chromosomes in *Figure 8* were straightened using the 'straighten-curved-objects' plugin in the Image J software (*Kocsis et al., 1991*).

## 5-methylcytosine (5mC) immunodetection

For immunodetection of 5mC, chromosome spreads were prepared according to the procedure described above. Denaturation mixture containing 20 µl of 50% formamide, 10% dextran sulfate in 2x SSC was pipetted onto each microscope slide. The slides were heated at 80°C for 2 min, washed in 2x SSC (2 × 5 min) and incubated in bovine serum albumin solution (5% BSA, 0.2% Tween-20 in 4x SSC) at 37°C for 30 min. Immunodetection was performed using 100 µl of primary antibody against 5mC(mouse anti 5mC, Diagenode, diluted 1: 100) at 37°C for 30 min. After washing in 2x SSC (2 × 5 min) the slides were incubated with the secondary antibody (Alexa Fluor 488 goat anti-mouse IgG, Invitrogen, diluted 1: 200) at 37°C for 30 min, followed by washing in 2x SSC (2 × 5 min) and a dehydration in an ethanol series (70%, 90%, and 100%, 2 min each). Chromosomes were counterstained with DAPI, fluorescence signals analyzed and photographed as described above. The slides were washed in 2x SSC (2 × 5 min), dehydrated in an ethanol series (70%, 90%, and 100%, 2 min each), and rehybridized with 35S rDNA probe as described above.

## Data availability

### Species resequencing

*A. barnebyi* (SRR7965809), *A. aurea* (SRR405095), *A. vulgaris* (SRR404349), *A. sibirica* (SRR405090), *A. formosa* (SRR408554), *A. japonica* (SRR413499), *A. oxysepala* (SRR413921), *A. longissima* (SRR7965810), *A. chrysantha* (SRR408559), *A. pubescens* (SRR7943924) are available in the Short Read Archive (https://www.ncbi.nlm.nih.gov/sra).

### Whole genome *Aquilegia coerulea* 'Goldsmith'

Sanger sequences used for genome assembly are available in the NCBI Trace Archive (https://www.ncbi.nlm.nih.gov/Traces)

### *Aquilegia coerulea* 'Goldsmith' ESTs

Available in the NCBI Short Read Archive (SRR505574-SRR505578)

### *Aquilegia formosa* 412 ESTs

Available in the NCBI dbEST (https://www.ncbi.nlm.nih.gov/dbEST/)

### *Aquilegia coerulea* 'Goldsmith' X *Aquilegia chrysantha* mapping population

Available in the NCBI Short Read Archive (SRR8000449-SRR8000976)

### *Aquilegia formosa* x *Aquilegia pubescens* mapping population

Available in the NCBI Short Read Archive (Bioproject PRJNA489508).

### grandparents

pub.1 (SRR7943925), pub.2 (SRR7943924), form.1 (SRR7790646), form.2 (SRR408554)

### F1s

F1.1 (SRR7943926), F1.2 (SRR7943927)

F2s

SRR7814612-SRR7814614, SRR7814616-SRR7814619, SRR7814622, SRR7814624-SRR7814686, SRR7826362-SRR7826624

RNAseq

Available in the NCBI Short Read Archive: see *Supplementary file 5* for more details

Other files

A vcf containing biallelic SNPs called in all ten *Aquilegia* species and *Semiaquilegia* (AQ.Semi.all.biallelic.SNPs.vcf.gz) and text files of genomic positions passing filtration (AQ.only.kept.positions.txt.gz and AQ.Semi.kept.positions.txt.gz) are available for download at dryad (doi:10.5061/dryad.j4j12v0).

URLs

The *A. coerulea* 'Goldsmith' v3.1 genome release is available at: https://phytozome.jgi.doe.gov/

## Acknowledgements

The authors would like to thank Daniel Rokhsar for guidance in initiating the study, Todd Perkins of Goldsmith Seeds (now part of Syngenta) for providing *A. coerulea* 'Goldsmith' seeds, and Gaku Kudo, Panayoti Kelaidis, and Len-Feng Li for wild collected seed. Next generation sequencing of some components of the *Aquilegia formosa* x *Aquilegia pubescens* mapping population was performed at the Vienna Biocenter Core Facilities Next Generation Sequencing (VBCF NGS) unit in Vienna, Austria (http://www.vbcf.ac.at). This work used the Vincent J Coates Genomics Sequencing Laboratory at UC Berkeley, supported by NIH S10 Instrumentation Grants S10RR029668 and S10RR027303. The work conducted by the U.S. Department of Energy Joint Genome Institute is supported by the Office of Science of the U.S. Department of Energy under Contract No. DE-AC02-05CH11231. TM and MAL were supported by the Czech Science Foundation (grant no. P501/12/G090) and the CEITEC 2020 (grant no. LQ1601) project. MK was supported by National Program of Sustainability I (grant no. LO1204). GA was supported by the Austrian Science Funds (FWF DK W1225-B20). ESB was supported by the UCSB Harvey L Karp Discovery Award and the National Institutes of Health under the Ruth L. Kirschstein National Research Service Award (F32GM103154). Work was also supported by NSF IOS 1456317 to SAH and NSF DEB 1311390 to SAH and NJD.

## Additional information

### Competing interests

Magnus Nordborg: Reviewing editor, *eLife*. The other authors declare that no competing interests exist.

### Funding

| Funder | Grant reference number | Author |
|---|---|---|
| University of California, Santa Barbara | Harvey L. Karp Discovery Award | Evangeline S Ballerini |
| National Institutes of Health | Ruth L. Kirschstein National Research Service Award F32GM103154 | Evangeline S Ballerini |
| Czech Science Foundation | P501/12/G090 | Terezie Mandáková Martin A Lysak |
| CEITEC 2020 | LQ1601 | Terezie Mandáková Martin A Lysak |
| Austrian Science Fund | FWF DK W1225-B20 | Gökçe Aköz |
| National Science Foundation | DEB 1311390 | Nathan J Derieg Scott A Hodges |

| National Program of Sustainability I | LO1204 | Miroslava Karafiátová |
| National Science Foundation | IOS 1456317 | Scott A Hodges |

The funders had no role in study design, data collection and interpretation, or the decision to submit the work for publication.

### Author contributions

Danièle L Filiault, Conceptualization, Data curation, Formal analysis, Investigation, Writing—original draft, Writing—review and editing, Analyzed species resequencing data, Performed population genetics; Evangeline S Ballerini, Data curation, Formal analysis, Investigation, Writing—review and editing, Performed F2 mapping, Provided RNAseq data; Terezie Mandáková, Martin A Lysak, Investigation, Writing—review and editing, Performed cytology; Gökçe Aköz, Formal analysis, Writing—original draft, Performed coalescent-based modelling; Nathan J Derieg, Formal analysis, Investigation, Writing—review and editing, Contributed RNAseq data and analysis; Jeremy Schmutz, Data curation, Supervision, Investigation, Writing—review and editing, Sequenced and assembled the reference; Jerry Jenkins, Jane Grimwood, Investigation, Writing—review and editing, Sequenced and assembled the reference; Shengqiang Shu, Richard D Hayes, Investigation, Writing—review and editing, Annotated the reference; Uffe Hellsten, Formal analysis, Investigation, Sequenced and assembled the reference; Kerrie Barry, Investigation, Project administration, Sequenced and assembled the reference; Juying Yan, Sirma Mihaltcheva, Viktoria Nizhynska, Investigation, Constructed and sequenced libraries; Miroslava Karafiátová, Investigation, Labeled the oligo paint library; Elena M Kramer, Conceptualization, Conceived the study, Provided RNAseq data; Scott A Hodges, Conceptualization, Supervision, Funding acquisition, Writing—original draft, Project administration, Writing—review and editing, Conceived the study; Magnus Nordborg, Conceptualization, Formal analysis, Supervision, Funding acquisition, Writing—original draft, Project administration, Writing—review and editing, Conceived the study

### Author ORCIDs

Danièle L Filiault  http://orcid.org/0000-0002-2938-3071
Kerrie Barry  http://orcid.org/0000-0002-8999-6785
Magnus Nordborg  http://orcid.org/0000-0001-7178-9748

### Decision letter and Author response

Decision letter https://doi.org/10.7554/eLife.36426.049
Author response https://doi.org/10.7554/eLife.36426.050

# Additional files

### Supplementary files

• Supplementary file 1. Genomic libraries included in the *A. coerulea* genome assembly and their respective assembled sequence coverage levels in the *A. coerulea* v3.1 release
DOI: https://doi.org/10.7554/eLife.36426.027

• Supplementary file 2. Summary statistics of the output of the whole genome shotgun assembly prior to screening, removal of organelles and contaminating scaffolds and chromosome-scale pseudomolecule construction
DOI: https://doi.org/10.7554/eLife.36426.028

• Supplementary file 3. Final summary assembly statistics for chromosome-scale assembly
DOI: https://doi.org/10.7554/eLife.36426.029

• Supplementary file 4. Placement of the individual BAC clones and their contribution to the overall error rate
DOI: https://doi.org/10.7554/eLife.36426.030

• Supplementary file 5. RNAseq data sets used for gene annotation
DOI: https://doi.org/10.7554/eLife.36426.031

- Supplementary file 6. Ratio of polymorphism or divergence on each chromosome versus genome-wide for each species
DOI: https://doi.org/10.7554/eLife.36426.032

- Supplementary file 7. Robustness of nucleotide diversity patterns to copy number variant detection methods
DOI: https://doi.org/10.7554/eLife.36426.033

- Supplementary file 8. Repeat family prevalence and permutation results in the *A. coerulea* v3.1 genome release
DOI: https://doi.org/10.7554/eLife.36426.034

- Supplementary file 9. *K*-mer based estimates of genome size and repetitive sequence proportion
DOI: https://doi.org/10.7554/eLife.36426.035

- Supplementary file 10. Mean and median coverage by species
DOI: https://doi.org/10.7554/eLife.36426.036

- Supplementary file 11. Proportion of sites removed by each filter - initial filtration without *Semiaquilegia*
DOI: https://doi.org/10.7554/eLife.36426.037

- Supplementary file 12. Proportion of sites removed by each filter - final filtration with *Semiaquilegia*
DOI: https://doi.org/10.7554/eLife.36426.038

- Supplementary file 13. Number of derived variants by species
DOI: https://doi.org/10.7554/eLife.36426.039

- Supplementary file 14. Transition matrix for the Five-State Markov process
DOI: https://doi.org/10.7554/eLife.36426.040

- Supplementary file 15. Transition matrix for the Eight-State Markov process
DOI: https://doi.org/10.7554/eLife.36426.041

## Data availability

Species resequencing: *A. barnebyi* (SRR7965809), *A. aurea* (SRR405095), *A. vulgaris* (SRR404349), *A. sibirica* (SRR405090), *A. formosa* (SRR408554), *A. japonica* (SRR413499), *A. oxysepala* (SRR413921), *A. longissima* (SRR7965810), *A. chrysantha* (SRR408559), *A. pubescens* (SRR7943924) are available in the Short Read Archive (https://www.ncbi.nlm.nih.gov/sra). Whole genome *Aquilegia coerulea* 'Goldsmith': Sanger sequences used for genome assembly are available in the NCBI Trace Archive (https://www.ncbi.nlm.nih.gov/Traces). *Aquilegia coerulea* 'Goldsmith' ESTs: Available in the NCBI Short Read Archive (SRR505574-SRR505578). *Aquilegia formosa* 412 ESTs: Available in the NCBI dbEST (https://www.ncbi.nlm.nih.gov/dbEST/). *Aquilegia coerulea* 'Goldsmith' X *Aquilegia chrysantha* mapping population: Available in the NCBI Short Read Archive (SRR8000449-SRR8000976). *Aquilegia formosa* x *Aquilegia pubescens* mapping population: Available in the NCBI Short Read Archive (BioProject PRJNA489508 [grandparents: pub.1 (SRR7943925), pub.2 (SRR7943924), form.1 (SRR7790646), form.2 (SRR408554); F1s: F1.1 (SRR7943926), F1.2 (SRR7943927); F2s: SRR7814612-SRR7814614, SRR7814616-SRR7814619, SRR7814622, SRR7814624-SRR7814686, SRR7826362-SRR7826624]). RNAseq: Available in the NCBI Short Read Archive (BioProject PRJNA490755; see Supplementary Table 5 for more details. Other files: A vcf containing biallelic SNPs called in all ten *Aquilegia* species and *Semiaquilegia* (AQ.Semi.all.biallelic.SNPs.vcf.gz) and text files of genomic positions passing filtration (AQ.only.kept.positions.txt.gz and AQ.Semi.kept.positions.txt.gz) are available for download at Dryad (doi:10.5061/dryad.j4j12v0). The *A. coerulea* 'Goldsmith' v3.1 genome release is available at: https://phytozome.jgi.doe.gov/.

The following datasets were generated:

| Author(s) | Year | Dataset title | Dataset URL | Database and Identifier |
|---|---|---|---|---|
| Filiault D, Ballerini ES, Mandakova T, Aköz G, Derieg N, Schmutz J, Jenkins J, Grimwood J, Shu S, Hayes RD, Hell- | 2018 | Genetic mapping of Aquilegia formosa and A. pubescens | https://www.ncbi.nlm.nih.gov/bioproject?term=PRJNA489508 | NCBI BioProject, PRJNA489508 |

| | | | | |
|---|---|---|---|---|
| sten U, Barry K, Yan J, Mihaltcheva S, Karafiatova M, Nizhynska V, Kramer EM, Lysak MA, Hodges SA, Nordborg M | | | | |
| Filiault D, Ballerini ES, Mandakova T, Aköz G, Derieg N, Schmutz J, Jenkins J, Grimwood J, Shu S, Hayes RD, Hellsten U, Barry K, Yan J, Mihaltcheva S, Karafiatova M, Nizhynska V, Kramer EM, Lysak MA, Hodges SA, Nordborg M | 2018 | Multiple Aquilegia species Transcriptome | https://www.ncbi.nlm.nih.gov/bioproject/?term=PRJNA490755 | NCBI BioProject, PRJNA490755 |
| Filiault D, Ballerini ES, Mandakova T, Aköz G, Derieg N, Schmutz J, Jenkins J, Grimwood J, Shu S, Hayes RD, Hellsten U, Barry K, Yan J, Mihaltcheva S, Karafiatova M, Nizhynska V, Lysak MA, Hodges SA, Nordborg M | 2018 | Data from: The Aquilegia genome provides insight into adaptive radiation and reveals an extraordinarily polymorphic chromosome with a unique history | https://dx.doi.org/10.5061/dryad.j4j12v0 | Dryad Digital Repository, 10.5061/dryad.j4j12v0 |

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
