## [Decision Letter]

Thank you for submitting your article "The *Aquilegia* genome: adaptive radiation and an extraordinarily polymorphic chromosome with a unique history" for consideration by *eLife*. Your article has been reviewed by Christian Hardtke as the Senior Editor, a Reviewing Editor, and three reviewers.

The following individual involved in review of your submission has agreed to reveal his identity: Andrew Leitch (Reviewer #3).

The reviewers have discussed the reviews with one another and the Reviewing Editor has drafted this decision to help you prepare a revised submission.

Summary:

This work presents an analysis of genomic variation within and among members of the *Aquilegia* genus, showing a complex pattern of allele sharing, population differentiation and remarkable heterogeneity across the genome, with one chromosome having an almost entirely different structure of diversity. These are useful data points in the effort to understand species radiation.

Essential revisions:

The paper is generally clear and well written. However, all reviewers felt that the connections between the analyses could be made a little clearer, that some statements were not well substantiated (e.g. see comments about background selection). I have included the individual reviews below so that you can see the points raised. I do not expect a full, point by point response to all comments, rather I ask that you revise to maximise coherence, rigour and perhaps also to identify future directions, given the descriptive nature of the work.

*Reviewer #1:*

This study appears to have started as a phylogenetic analysis of several *Aquilegia* species, and seems to have uncovered an odd finding, which the present manuscript describes, but cannot interpret with much confidence. The species are close relatives, and many variants are shared between them, even though the samples are very small, with most species represented by a single individual.

The observation that diploid *Aquilegia* species have one chromosome that differs cytologically from the other 6 pairs dates back to a paper describing pachytene chromosomes of a species different from the ones in this study, *A. glandulosa*, that was published in 1961, and that the manuscript cites (albeit with a typo in the journal name, which should be "Chromosoma"). This chromosome carries the nucleolar organiser and has 2 large heterochromatic blocks either side of it, that resemble the pericentromeric regions of the other chromosomes. The region appears to be rearranged in *A. longissima*, as the hybrids exhibit a complex arrangement, apparently a quadrivalent. The 1961 paper suggests that rearrangements may be promoted by what would now be called ectopic pairing between the two repetitive regions associated with the nucleolar organizers on the 2 arms, and that the intervening region would show a pericentric inversion, or at least be forced to pair with non-homologous regions, inhibiting crossing over, and promoting rearrangements. Ectopic pairing is supported by observations of quadrivalents. Even within *A. glandulosa*, the arrangement of this chromosome appears to be polymorphic, again with observations, such as "foldbacks" suggesting ectopic pairing.

However, in this ms, although one chromosome (4) exhibits differences from the others (see below), subsection “The 35S and 5S rDNA loci are uniquely localized to chromosome four” states that it is not more heterochromatic than other chromosomes. It would be good to be clear about whether the old observations are now in doubt. If so, it seems odd to suggest that rDNA clusters may be involved (though the text at the end of this page is very vague, so perhaps it is not intended to support Linnert's hypotheses.

The haploid genome size is not extraordinarily high (around 500 Mb). This is about twice the assembly size in the manuscript, which does not mention the estimated total DNA content. Part of the discrepancy may therefore be because only sequences with EST coverage above 85% were retained, and it is unclear what tissue was used for the EST sequencing, and whether it could lack a substantial proportion of the genomic transcripts, or perhaps because the high contamination with arthropod and prokaryote ESTs led to current information being very incomplete. If this can be excluded, it would suggest that this "odd" chromosome has accumulated large amounts of repetitive sequences, presumably because crossing over is infrequent. It is unclear why this might be the case, but the results suggest that some factor has led to fairly long-term balancing selection, since different species share variants, and presumably also share different arrangements.

The manuscript attempts to interpret the findings in terms of weak purifying selection, presumably allowing high diversity, so that variants might be shared between species. Another alternative is balancing selection, as the text mentions. However, it is misleading to say that the chromosome with the odd behavior (chromosome four) "is enriched for defense genes, some of which have been shown to be under balancing selection", without mentioning that this refers to other species (which are not specified in the text; it should also be explained what kind of defense is meant). The enrichment is said to be small, without explicit details. The type of balancing selection is also left vague. It seems unlikely to involve heterozygote advantage if the species are indeed inbreeders (though this is not entirely clear, see comments below). The idea of a "supergene" is mentioned, but not explained. Is such a situation plausible in this case? Could local adaptation be involved?

Another possibility, background selection, is discussed, again without any cogent analysis to test it. Low gene density is mentioned for this chromosome, but readers are referred to a Supplementary file. If this is important information, it should be in the main text. Is anything known about recombination? The manuscript mentions genetic maps that were used to assist the genome assembly, so perhaps they can be helpful?

The comparison of Fst values in subsection “Polymorphism and divergence” seems very odd. FST between (presumably geographic) regions was high (0.245-0.271) was lower than between most vervet species pairs, but higher than between cichlid groups in Malawi, or human ethnic groups. Is this correct? I am not an expert on human genetics, but my impression was that Fst between different human populations was nearer 5-10%. For the cichlids, "groups" is a very vague word, and of course different values of Fst may reflect different processes, including the possibility that some "groups" have low within-group diversity due to recent population bottlenecks. For the vervets, the value presumably depends mainly on the divergence between the specific species included in the study. The phrase "between common Arabidopsis species" is so vague that it cannot be understood, but clearly some species in this genus are inbreeders, with documented low diversity (and thus high Fst between samples). Why not compare with other plant species for which there are selfing rate estimates that suggest a similar breeding system? Fst values can be based on non-SNP data, such as allozymes, and large compilations of estimates have been published, e.g. Hamrick and Godt, (1996).

The Discussion section consists mainly of repeating the results. It would be better to test some hypotheses and discuss the implications of the tests.

*Reviewer #2:*

Fillault et al., summarize intriguing results of population and speciation genomic variation in columbines. They provide two major results: (1) A large number of topologies across the genome, which reflect both incomplete lineage sorting and introgression (2) A pattern of elevated diversity and introgression on chromosome 4, a result that likely reflect some escape from the action of linked selection on this chromosome. On the whole the results are quite interesting, however, the paper does not cohere as much as I would like. Here I provide suggestions by which the paper could be improved. It is not clear to me if these changes would result in a product appropriate for *eLife*, but they could…

Writing: The writing is serviceable, but the urgency of the questions and the relationship between the two findings does not shine through. The observations of introgression, while interesting on their own, do not provide a major result at this time. The observation on chromosome 4, one the other had does. This paper could be strengthened by making the testing for a connection between these observations (see suggestion below).

More broadly a serious re-write and refocus is necessary. For example, the final paragraph seems to be strangely placed etc..

Additional analyses:

1) It would be worthwhile to test if local topology (or perhaps D stat) is predictable by gene density or some such measure of linked selection. If so, this would be a nice way to connect the two results. I see that the authors did not find a relationship between gene density and diversity genome wide, but I did not see a corresponding analysis of gene density and introgression. If such a result were observed it would strengthen the paper by providing a unified explanation for differences in introgression rates on Chr4 vs genome wide patterns AND for local variation in introgression across the genome.

2) Regardless, the Chr4 result is quite novel, and will likely be more impactful than the introgression result, so a change in structure and focus could help. It would for example, be interesting to estimate the genome-wide strength of linked selection relative to Chr4 as a best guess of no linked selection.

Citations etc:

The description of "D" stats is somewhat idiosyncratic and does not cite any of the literature on these approaches. Similarly, the discussion of linked selection should refer to Corbet-Detig's recent paper (see also point 2, above).

*Reviewer #3:*

This is a very interesting paper, with lots of high quality analyses, expertly performed, including a very nice mixture of population genetics and genomics. The authors discover, amongst 10 of 70 *Aquelegia* species, a group that has recently radiated with have high levels of allele sharing. The authors also infer the origin of an ancestral chromosome 4, that has unusually high levels of polymorphisms and low gene densities, suggestive of a chromosome that is not shared so widely as other regions of the genome. They suggest that patterns of polymorphisms in this chromosome 4 are unusual, and perhaps unique. Collectively this manuscript represents a substantial body of work, leading to new insights in genome evolution and into *Aquelegia* speciation processes.

Levels of polymorphisms are compared between the 7 *Aquelegia* chromosomes they reconstruct. They are derived from resequence data from single individuals of each species analysed. The measures are from genes and repeats alike, which are not unpicked. Chromosome 4 provides a central thrust to the manuscript's interest. This chromosome has a low gene density, high repeat density and high levels of polymorphisms.

Given chromosome 4's importance, the manuscript would benefit from additional text in the Results section to supplement "With the aid of genetic maps, we assembled sequences into a 291.7 Mb reference genome consisting of 7 chromosomes (282.6 Mbp) and an additional 1,027 scaffolds". This will help explain how these chromosomes were reconstructed and to briefly explain how sequence scaffolds, linkage maps and BACs were used to assemble the chromosomes for analysis. (The descriptions are detailed in M and M).

The features of chromosome 4 may be unique, and are compared with sex chromosomes. Perhaps too they could be compared with B chromosomes. Could chromosome 4 be a balanced pair of ancestral B chromosomes, now fixed in Aquilegia? High levels of polymorphisms and repeats might support such a hypothesis, and the accumulation of adaptive genes (i.e the high fraction of defense genes they observe) could then prevent them from introgressing between species…rDNA loci are also known on B chromosomes, as are pseudogenes and zero and low expression of chromosome 4 genes is also observed)… this hypothesis represent only a suggestion, for the authors to consider (and reject if they do not like it).

The authors note unusual pairing patterns (variable bubbles and fold backs) for 35S rDNA on chromosome 4. Could these structures simply represent delayed synapsis, or early diplotene phase, perhaps associated with NOR activity…certainly I see no such structures in meiocytes in supplementary materials? Is the boxed bivalent on Figure 7 from another meiocyte, it needs to be made clear.

The Abstract is very brief and could be expanded to better reflect the interest of the paper. "Sheds light on complex process of adaptive radiation": what light is that, it should be spelled out? The Abstract and Figure 1 also points out that the species studied have a wide geographic distribution. Whilst that is interesting, perhaps more useful would be descriptions of the phylogenetic distribution in the genus, how representative are these species of the diversity across the genus?

---

## [Author Response]

Many thanks for the reviews and the decision. As might be inferred from the time it has taken us to revise this, we took the comments quite seriously, and believe we have significantly improved the manuscript. As requested, we do not provide a point-by-point response, but note that we have expanded and clarified all relevant sections, and carried out the analyses suggested by reviewer #2 (see new Figure 7). Indeed, we find that there is a relationship between gene flow and gene density (and refer to the recent paper from Molly Schumer), but we also find that gene density alone cannot explain the unprecedented differences in polymorphism level between chromosomes. What explains this will be the focus of future work (now mentioned in the Discussion section).